# Fast and efficient MATLAB-based MPM solver (fMPMM-solver v1.1)

Emmanuel Wyser[1], Yury Alkhimenkov[1,2,3], Michel Jaboyedoff[1,2], and Yury Y. Podladchikov[1,2,3]

[1]Institute of Earth Sciences, University of Lausanne, 1015 Lausanne, Switzerland
[2]Swiss Geocomputing Centre, University of Lausanne, 1015 Lausanne, Switzerland
[3]Department of Mechanics and Mathematics, Moscow State University, Moscow, Russia

**Correspondence:** Emmanuel Wyser (manuwyser@gmail.com)

**Abstract.** We present an efficient MATLAB-based implementation of the material point method (MPM) and its most recent variants. MPM has gained popularity over the last decade, especially for problems in solid mechanics in which large deformations are involved, i.e., cantilever beam problems, granular collapses, and even large-scale snow avalanches. Although its numerical accuracy is lower than that of the widely accepted finite element method (FEM), MPM has been proven useful in overcoming some of the limitations of FEM, such as excessive mesh distortions. We demonstrate that MATLAB is an efficient high-level language for MPM implementations that solve elasto-dynamic and elasto-plastic problems. We accelerate the MATLAB-based implementation of MPM method by using the numerical techniques recently developed for FEM optimization in MATLAB. These techniques include vectorisation, the usage of native MATLAB functions, the maintenance of optimal RAM-to-cache communication, and others. We validate our in-house code with classical MPM benchmarks including i) the elastic collapse under self-weight of a column, ii) the elastic cantilever beam problem, and iii) existing experimental and numerical results, i.e., granular collapses and slumping mechanics respectively. We report a performance gain by a factor of 28 for a vectorised code compared to a classical iterative version. The computational performance of the solver is at least 2.8 times greater than those of previously reported MPM implementations in Julia under a similar computational architecture.

## 1   Introduction

The material point method (MPM), developed in the 1990s (Sulsky et al., 1994), is an extension of a particle-in-cell (PIC) method to solve solid mechanics problems involving massive deformations. It is an alternative to Lagrangian approaches (updated Lagrangian finite element method) that is well suited to problems with large deformations involved in geomechanics, granular mechanics or even snow avalanche mechanics. Vardon et al. (2017); Wang et al. (2016c) investigated elasto-plastic problems of strain localization of slumping processes relying on an explicit or implicit MPM formulation. Similarly, Bandara et al. (2016); Bandara and Soga (2015); Abe et al. (2014) proposed a poro-elasto-plastic MPM formulation to study levee failures induced by pore pressure increases. Additionally, Baumgarten and Kamrin (2019); Dunatunga and Kamrin (2017, 2015); Więckowski (2004) proposed a general numerical framework of granular mechanics, i.e., silo discharge or granular collapses. More recently, Gaume et al. (2019, 2018) proposed a unified numerical model in the finite deformation framework to study the whole process, i.e., from failure to propagation, of slab avalanche releases.

The core idea of MPM is to discretize a continuum with material points carrying state variables (e.g., mass, stress, and velocity). The latter are mapped (accumulated) to the nodes of a regular or irregular background FE mesh, on which an Eulerian solution to the momentum balance equation is explicitly advanced forward in time. Nodal solutions are then mapped back to the material points, and the mesh can be discarded. The mapping from material points to nodes is ensured using the standard FE hat function that spans over an entire element (Bardenhagen and Kober, 2004). This avoids a common flaw of 30 FEM, which is an excessive mesh distortion. We will refer to this first variant as the standard material point method (sMPM).

  MATLAB$^{©}$ allows a rapid code prototyping but, at the expense of significantly lower computational performances than compiled language. An efficient MATLAB implementation of FEM called MILAMIN (Million a Minute) was proposed by Dabrowski et al. (2008) that was capable of solving two-dimensional linear problems with one million unknowns in one minute on a modern computer with a reasonable architecture. The efficiency of the algorithm lies on a combined use of vectorised 35 calculations with a technique called blocking. MATLAB uses the Linear Algebra PACKages (LAPACK), written in Fortran, to perform mathematical operations by calling Basic Linear Algebra Subroutines (BLAS, Moler 2000). The latter results in an overhead each time a BLAS call is made. Hence, mathematical operations over a large number of small matrices should be avoided and, operations on fewer and larger matrices preferred. This is a typical bottleneck in FEM when local stiffness matrices are assembled during the integration point loop within the global stiffness matrix. Dabrowski et al. (2008) proposed an 40 algorithm, in which a loop reordering is combined with operations on blocks of elements to address this bottleneck. However, data required for a calculation within a block should entirely reside in the CPU's cache. Otherwise, an additional time is spent on the RAM-to-cache communication and the performance decreases. Therefore, an optimal block size exists and, is solely defined by the CPU architecture. This technique of vectorisation combined with blocking significantly increases the performance.

More recently, Bird et al. (2017) extended the vectorised and blocked algorithm presented by Dabrowski et al. (2008) to the calculation of the global stiffness matrix for Discontinuous Galerkin FEM considering linear elastic problems using only native MATLAB functions. Indeed, the optimisation strategy chosen by Dabrowski et al. (2008) also relied on non-native MATLAB functions, e.g., `sparse2` of the SuiteSparse package (Davis, 2013). In particular, Bird et al. (2017) showed the importance of storing vectors in a column-major form during calculation. Mathematical operations are performed in MATLAB by calling 50 LAPACK, written in FORTRAN, in which arrays are stored in column-major order form. Hence, element-wise multiplication of arrays in column-major form is significantly faster and thus, vectors in column-major form are recommended, whenever possible. Bird et al. (2017) concluded that vectorisation alone results in a performance increase between 13.7 and 23 times, while blocking only improved vectorisation by an additional 1.8 times. O'Sullivan et al. (2019) recently extended the works of Bird et al. (2017); Dabrowski et al. (2008) to optimised elasto-plastic codes for Continuous Galerkin (CG) or Discontin-55 uous Galerkin (DG) methods. In particular, they proposed an efficient native MATLAB function, i.e., `accumarray()`, to efficiently assemble the internal force vector. Such function constructs an array by accumulation. More generally, O'Sullivan et al. (2019) reported a performance gain of x25.7 when using an optimised CG code instead of an equivalent non-optimised code.

Since MPM and FEM are similar in their structure, we aim at increasing the performances of MATLAB up to what was reported by Sinaie et al. (2017) using Julia language environment. In principal, Julia is significantly faster than MATLAB for a MPM implementation. We combine the most recent and accurate versions of MPM: the explicit generalized interpolation material point method (GIMPM, Bardenhagen and Kober 2004) and the explicit convected particle domain interpolation with second-order quadrilateral domains (CPDI2q and CPDI, Sadeghirad et al. 2013, 2011) variants with some of the numerical techniques developed during the last decade of FEM optimisation in MATLAB. These techniques include the use of `accumarray()`, optimal RAM-to-cache communication, minimum BLAS calls and the use of native MATLAB functions. We did not consider the blocking technique initially proposed by Dabrowski et al. (2008) since an explicit formulation in MPM excludes the global stiffness matrix assembly procedure. The performance gain mainly comes from the vectorisation of the algorithm, whereas blocking has a less significant impact over the performance gain, as stated by Bird et al. (2017). The vectorisation of MATLAB functions is also crucial for a straight transpose of the solver to a more efficient language, such as the C-CUDA language, which allows the parallel execution of computational kernels of graphics processing units (GPUs).

In this contribution, we present an implementation of an efficiently vectorised explicit MPM solver (fMPMM-solver, which v1.1 is available for download from Bitbucket at: https://bitbucket.org/ewyser/fmpmm-solver/src/master/), taking advantage of vectorisation capabilities of MATLAB$^{©}$. We extensively use native functions of MATLAB$^{©}$ such as `repmat( )`, `reshape( )`, `sum( )` or `accumarray( )`. We validate our in-house code with classical MPM benchmarks including i) the elastic collapse under self-weight of a column, ii) the elastic cantilever beam problem, and iii) existing experimental results, i.e., granular collapses and slumping mechanics. We demonstrate the computational efficiency of a vectorised implementation over an iterative one for the case of an elasto-plastic collapse of a column. We compare the performances of Julia and MATLAB language environments for the collision of two elastic discs problem.

## 2  Overview of the Material Point Method (MPM)

### 2.1  A Material Point Method implementation

The material point method (MPM), originally proposed by Sulsky et al. (1995, 1994) in an explicit formulation, is an extension of the particle-in-cell (PIC) method. The key idea is to solve the weak form of the momentum balance equation on a FE mesh while state variables (e.g., stress, velocity or mass) are stored at Lagrangian points discretizing the continuum, i.e., the material points, which can move according to the deformation of the grid (Dunatunga and Kamrin, 2017). MPM could be regarded as a finite element solver in which integration points (material points) are allowed to move (Guilkey and Weiss, 2003) and are thus not always located at the Gauss-Legendre location within an element, resulting in higher quadrature errors and poorer integration estimates, especially when using low-order basis functions (Steffen et al., 2008a, b).

A typical calculation cycle (see Fig. 1) consists of the three following steps (Wang et al., 2016a):

  1. A Mapping phase, during which properties of the material point (mass, momentum or stress) are mapped to the nodes.

2. An updated-Lagrangian FEM (UL-FEM) phase, during which the momentum equations are solved on the nodes of the background mesh and, the solution is explicitly advanced forward in time.

3. A Convection phase, during which i) the nodal solutions are interpolated back to the material points, and ii) the properties of the material point are updated.

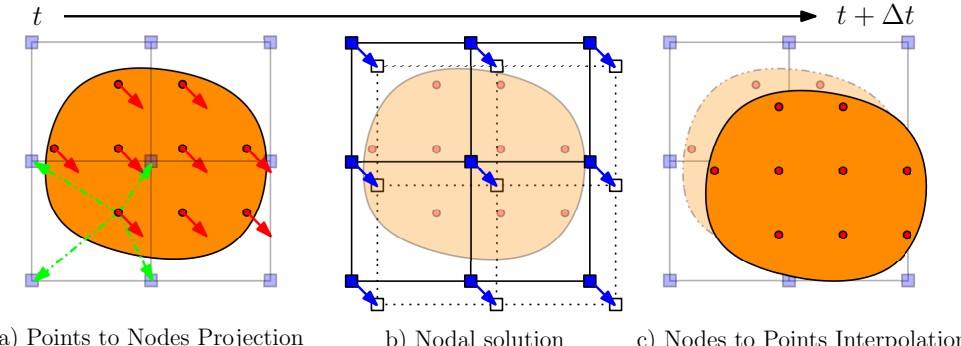

a) Points to Nodes Projection   b) Nodal solution   c) Nodes to Points Interpolation

**Figure 1.** Typical calculation cycle of a MPM solver for a homogeneous velocity field, inspired by Dunatunga and Kamrin (2017). a) The continuum (orange) is discretized into a set of Lagrangian material points (red dots), at which state variables or properties (e.g., mass, stress, and velocity) are defined. The latter are mapped to an Eulerian finite element mesh made of nodes (blue square). b) Momentum equations are solved at the nodes and, the solution is explicitly advanced forward in time. c) The nodal solutions are interpolated back to the material points and, their properties are updated.

Since the 1990's, several variants were introduced to resolve a number of numerical issues. The generalized interpolation material point method (GIMPM) was first presented by Bardenhagen and Kober (2004). They proposed a generalization of the basis and gradient functions that were convoluted with a characteristic domain function of the material point. A major flaw in sMPM is the lack of continuity of the gradient basis function, resulting in spurious oscillations of internal forces as soon as a material point crosses an element boundary while entering into its neighbour. This is referred to as cell-crossing instabilities due to the $C_0$ continuity of the gradient basis functions used in sMPM. Such issue is minimized by the GIMPM variant (Acosta et al., 2020).

GIMPM is categorized as a domain-based material point method, unlike the later development of the B-spline material point method (BSMPM, e.g. de Koster et al. 2020; Gan et al. 2018; Gaume et al. 2018; Stomakhin et al. 2013) which cures cell-crossing instabilities using B-spline functions as basis functions. Whereas in sMPM only nodes belonging to an element contribute to a given material point, GIMPM requires an extended nodal connectivity, i.e., the nodes of the element enclosing the material point and the nodes belonging to the adjacent elements (see Fig. 2). More recently, the convected particle domain interpolation (CPDI and its most recent development CPDI2q) has been proposed by Sadeghirad et al. (2013, 2011).

We choose the explicit GIMPM variant with the modified update stress last scheme (MUSL, see Nairn (2003); Bardenhagen et al. (2000) for a detailed discussion), i.e., the stress of material point is updated after the nodal solutions are obtained. The updated momentum of the material point is then mapped back a second time to the nodes in order to obtain an updated nodal

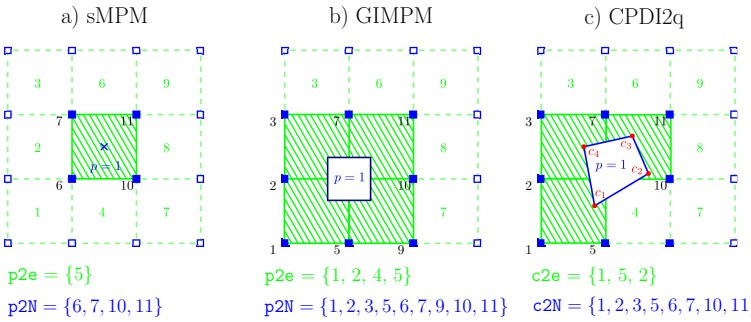

**Figure 2.** Nodal connectivities of a) standard MPM, b) GIMPM and c) CPDI2q variants. The material point's location is marked by the blue cross. Note that for sMPM (and similarly BSMPM) the particle domain does not exist, unlike GIMPM or CPDI2q (the blue square enclosing the material point). Nodes associated with the material point are denoted by filled blue squares, and the element number appears in green in the centre of the element. For sMPM and GIMPM, the connectivity array between the material point and the element is p2e and, the array between the material point and its associated nodes is p2N. For CPDI2q, the connectivity array between the corners (filled red circles) of the quadrilateral domain of the material point and the element is c2e and, the array between the corners and their associated nodes is c2N.

velocity, further used to calculate derivative terms such as strains or the deformation gradient of the material point. The explicit formulation also implies the well-known restriction on the time step, which is limited by the Courant-Friedrich-Lewy (CFL) condition to ensure numerical stability.

Additionally, we implemented a CPDI/CPDI2q version (in an explicit and quasi-static implicit formulation) of the solver. However, in this paper, we do not present the theoretical background of the CPDI variant nor the implicit implementation of a MPM-based solver. Therefore, interested readers are referred to the original contributions of Sadeghirad et al. (2013, 2011) and Acosta et al. (2020); Charlton et al. (2017); Iaconeta et al. (2017); Beuth et al. (2008); Guilkey and Weiss (2003), respectively. Regarding the quasi-static implicit implementation, we strongly adapted our vectorisation strategy to some aspects of the numerical implementation proposed by Coombs and Augarde (2020) in the MATLAB code AMPLE v1.0. However, we did not consider blocking, because our main concern for performance is on the explicit implementation.

## 2.2 Domain-based material point method variants

Domain-based material point method variants could be treated as two distinct groups:

- The material point's domain is a square for which the deformation is always aligned with the mesh axis, i.e., a non-deforming domain uGIMPM (Bardenhagen and Kober, 2004) or, a deforming domain cpGIMPM (Wallstedt and Guilkey, 2008), the latter being usually related to a measure of the deformation, e.g., the determinant of the deformation gradient.

- The material point's domain is either a deforming parallelogram for which its dimensions are specified by two vectors, i.e., CPDI (Sadeghirad et al., 2011), or a deforming quadrilateral solely defined by its corners, i.e., CPDI2q (Sadeghirad et al., 2013). However, the deformation is not necessarily aligned with the mesh anymore.

We first focus on the different domain updating methods for GIMPM. Four domain updating methods exists: i) the domain is not updated, ii) the deformation of the domain is proportional to the determinant of the deformation gradient $\det(F_{ij})$ (Bardenhagen and Kober, 2004), iii) the domain lengths $l_p$ are updated accordingly to the principal component of the deformation gradient $F_{ii}$ (Sadeghirad et al., 2011) or, iv) are updated with the principal component of the stretch part of the deformation gradient $U_{ii}$ (Charlton et al., 2017). Coombs et al. (2020) highlighted the suitability of generalised interpolation domain updating methods accordingly to distinct deformation modes. Four different deformation modes were considered by Coombs et al. (2020): simple stretch, hydrostatic compression/extension, simple shear and, pure rotation. Coombs et al. (2020) concluded the following:

- Not updating the domain is not suitable for simple stretch and hydrostatic compression/extension.

- A domain update based on $\det(F_{ij})$ will results in an artificial contraction/expansion of the domain for simple stretch.

- The domain will vanish with increasing rotation when using $F_{ii}$.

- The domain volume will change under isochoric deformation when using $U_{ii}$.

Consequently, Coombs et al. (2020) proposed a hybrid domain update inspired by CPDI2q approaches: the corners of the material point domain are updated accordingly to the nodal deformation but, the midpoints of the domain limits are used to update domain lengths $l_p$ to maintain a rectangular domain. Even though Coombs et al. (2020) reported an excellent numerical stability, the drawback is to compute specific basis functions between nodes and material point's corners, which has an additional computational cost. Hence, we did not selected this approach in this contribution.

Regarding the recent CPDI/CPDI2q, Wang et al. (2019) investigated the numerical stability under stretch, shear and torsional deformation modes. CPDI2q was found to be erroneous in some case, especially when torsion mode is involved, due to distortion of the domain. In contrast, CPDI and even sMPM performed better in modelling torsional deformations. Even though CPDI2q can exactly represent the deformed domain (Sadeghirad et al., 2013), care must be taken when dealing with very large distortion, especially when the material has yielded, which is common in geotechnical engineering (Wang et al., 2019).

Consequently, the domain-based method as well as the domain updating method should be carefully chosen accordingly to the deformation mode expected for a given case. The latter will be always specify in the following and, the domain update method will be clearly stated.

## 3 MATLAB-based MPM implementation

### 3.1 Rate formulation and elasto-plasticity

The large deformation framework in a linear elastic continuum requires an appropriate stress-strain formulation. One approach is based on the finite deformation framework, which relies on a linear relationship between elastic logarithmic strains and

Kirchoff stresses (Coombs et al., 2020; Gaume et al., 2018; Charlton et al., 2017). In this study, we adopt another approach, namely, a rate dependent formulation using the Jaumann stress rate (e.g. Huang et al. 2015; Bandara et al. 2016; Wang et al. 2016c, b). This formulation provides an objective (invariant by rotation or frame-indifferent) stress rate measure (de Souza Neto et al., 2011) and is simple to implement. The Jaumann rate of the Cauchy stress is defined as

$$\frac{\mathcal{D}\sigma_{ij}}{\mathcal{D}t} = \frac{1}{2}C_{ijkl}\left(\frac{\partial v_l}{\partial x_k} + \frac{\partial v_k}{\partial x_l}\right), \tag{1}$$

where $C_{ijkl}$ is the fourth rank tangent stiffness tensor and $v_k$ is the velocity. Thus, the Jaumann stress derivative can be written as

$$\frac{\mathcal{D}\sigma_{ij}}{\mathcal{D}t} = \frac{\mathrm{D}\sigma_{ij}}{\mathrm{D}t} - \sigma_{ik}\omega_{jk} - \sigma_{jk}\omega_{ik}, \tag{2}$$

where $\omega_{ij} = (\partial_i v_j - \partial_j v_i)/2$ is the vorticity tensor and $\mathrm{D}\sigma_{ij}/\mathrm{D}t$ denotes the material derivative

$$\frac{\mathrm{D}\sigma_{ij}}{\mathrm{D}t} = \frac{\partial \sigma_{ij}}{\partial t} + v_k\frac{\partial \sigma_{ij}}{\partial x_k}. \tag{3}$$

Plastic deformation is modelled with a pressure dependent Mohr-Coulomb law with non-associated plastic flow, i.e., both the dilatancy angle $\psi$ and the volumetric plastic strain $\epsilon_v^p$ are null (Vermeer and De Borst, 1984). We have adopted the approach of Simpson (2017) for a two dimensional linear elastic, perfectly plastic (elasto-plasticity) continuum because of its simplicity and its ease of implementation. The yield function is defined as

$$f = \tau + \sigma\sin\phi - c\cos\phi, \tag{4}$$

where $c$ is the cohesion and $\phi$ the angle of internal friction,

$$\sigma = (\sigma_{xx} + \sigma_{yy})/2, \tag{5}$$

and

$$\tau = \sqrt{(\sigma_{xx} - \sigma_{yy})^2/4 + \sigma_{xy}^2}. \tag{6}$$

The elastic state is defined when $f < 0$. However when $f > 0$, plastic state is declared and stresses must be corrected (or scaled) to satisfy the condition $f = 0$, since $f > 0$ is an inadmissible state. Simpson (2017) proposed the following simple algorithm to return stresses to the yield surface,

$$\sigma_{xx}^\star = \sigma + (\sigma_{xx} - \sigma_{yy})\beta/2, \tag{7}$$

$$\sigma_{yy}^\star = \sigma - (\sigma_{xx} - \sigma_{yy})\beta/2, \tag{8}$$

$$\sigma_{xy}^\star = \sigma_{xy}\beta, \tag{9}$$

where $\beta = (| c \cos \phi - \sigma \sin \phi |)/\tau$, and $\sigma_{xx}^{\star}$, $\sigma_{yy}^{\star}$ and $\sigma_{xy}^{\star}$ are the corrected stresses, i.e., $f = 0$.

A similar approach is used to return stresses when considering a non-associated Drucker-Prager plasticity (see Huang et al. (2015) for a detailed description of the procedure). In addition, their approach allows also to model associated plastic flows, i.e., $\psi > 0$ and $\epsilon_v^p \neq 0$.

### 3.2 Structure of the MPM solver

The solver procedure is shown in Fig. 3. In the `main.m` script, both functions `meSetup.m` and `mpSetup.m`, respectively, define the geometry and related quantities such as the nodal connectivity (or element topology) array, e.g., the `e2N` array. The latter stores the nodes associated with a given element. As such, a material point $p$ located in an element $e$ can immediately identify which nodes $n$ it is associated with.

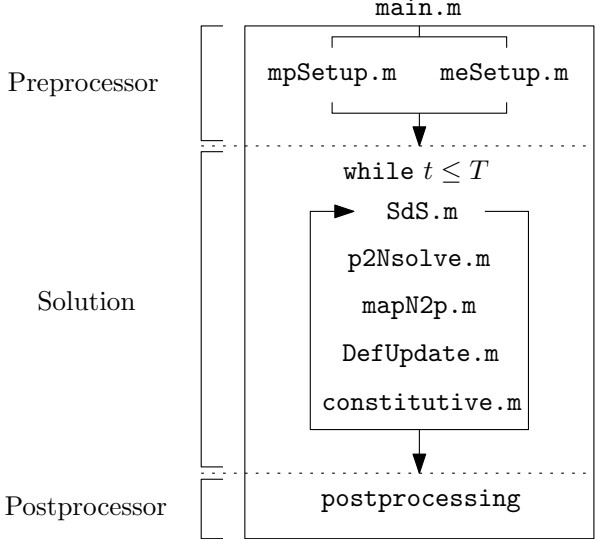

**Figure 3.** Workflow of the explicit GIMPM solver and the calls to functions within a calculation cycle. The role of each function is described in the text.

After initialization, a while loop solves the elasto-dynamic (or elasto-plastic) problem until a time criterion $T$ is reached.
This time criterion could be restricted to the time needed for the system to reach an equilibrium, or if the global kinetic energy of the system has reached a threshold.

At the beginning of each cycle, a connectivity array `p2e` between the material points and their respective element (a material point can only reside in a single element) is constructed. Since i) the nodes associated with the elements and ii) the elements enclosing the material points are known, it is possible to obtain the connectivity array `p2N` between the material points and
200 their associated nodes, e.g., `p2N=e2N(p2e,:)` in a MATLAB syntax (see Fig. 2 for an example of these connectivity arrays). This array is of dimension $(n_p, n_n)$, with $n_p$ the total number of material points, $n_n$ the total number of nodes associated with an element (16 in two-dimensional problems) and $n_{i,j}$ the node number where $i$ corresponds to the material point and $j$

corresponds to its j-th associated nodes, which results in the following:

$$\mathtt{p2N} = \begin{pmatrix} n_{1,1} & \cdots & n_{1,n_n} \\ \vdots & \ddots & \vdots \\ n_{n_p,1} & \cdots & n_{n_p,n_n} \end{pmatrix}. \tag{10}$$

The following functions are called successively during one calculation cycle:

1. `SdS.m` calculates the basis functions, derivatives and, assembles the strain-displacement matrix for each material points.

2. `p2Nsolve.m` projects the quantities of the material point (e.g., mass and momentum) to the associated nodes, solves the equations of motion and sets boundary conditions.

3. `mapN2p.m` interpolates nodal solutions (acceleration and velocity) to the material points with a double mapping procedure (see Zhang et al. (2016) or Nairn (2003) for a clear discussion of USF, USL and MUSL algorithms).

4. `DefUpdate.m` updates incremental strains and deformation-related quantities (e.g., the volume of the material point or the domain half-length) at the level of the material point based on the remapping of the updated material point momentum.

5. `constitutive.m` calls two functions to solve for the constitutive elasto-plastic relation, i.e.,

   (a) `elastic.m`, which predicts an incremental objective stress assuming a purely elastic step, further corrected by

   (b) `plastic.m`, which corrects the trial stress by a plastic correction if the material has yielded.

When a time criterion is met, the calculation cycle stops and further post-processing tasks (visualization, data exportation) can be performed.

The numerical simulations are conducted using MATLAB© R2018a within a Windows 7 64-bit environment on an Intel Core i7-4790 (4th generation CPU with 4 physical cores of base frequency at 3.60 GHz up to a maximum turbo frequency of 4.00 GHz) with $4 \times 256$ kB L2 cache and 16 GB DDR3 RAM (clock speed 800 MHz).

### 3.3 Vectorisation

#### 3.3.1 Basis functions and derivatives

The GIMPM basis function (Coombs et al., 2018; Steffen et al., 2008a; Bardenhagen and Kober, 2004) results from the convolution of a characteristic particle function $\chi_p$ (i.e., the material point spatial extent or domain) with the standard basis

function $N_n(x)$ of the mesh, which results in:

$$S_n(x_p) = \begin{cases} 1 - (4x^2 + l_p^2)/(4hl_p) & \text{if } |x| < l_p/2 \\ 1 - |x|/h & \text{if } l_p/2 \leq |x| < h - l_p/2 \\ (h + l_p/2 - |x|)^2/(2hl_p) & \text{if } h - l_p/2 \leq |x| < h + l_p/2 \\ 0 & \text{otherwise}, \end{cases} \tag{11}$$

with $l_p$ the length of the material point domain, $h$ the mesh spacing, $x = x_p - x_n$ where $x_p$ is the coordinate of a material point and $x_n$ the coordinate of its associated node $n$. The basis function of a node $n$ with its material point $p$ is constructed for a two-dimensional model, as follows:

$$S_n(\boldsymbol{x}_p) = S_n(x_p)S_n(y_p), \tag{12}$$

for which the derivative is defined as:

$$\nabla S_n(\boldsymbol{x}_p) = (\partial_x S_n(x_p)S_n(y_p), S_n(x_p)\partial_y S_n(y_p)). \tag{13}$$

Similar to the FEM, the strain-displacement matrix $\boldsymbol{B}$ consists of the derivatives of the basis function and is assigned to each material point, which results in the following:

$$\boldsymbol{B}(\boldsymbol{x}_p) = \begin{pmatrix} \partial_x S_1 & 0 & \cdots & \partial_x S_{n_n} & 0 \\ 0 & \partial_y S_1 & \cdots & 0 & \partial_y S_{n_n} \\ \partial_y S_1 & \partial_x S_1 & \cdots & \partial_y S_{n_n} & \partial_x S_{n_n} \end{pmatrix}, \tag{14}$$

where $n_n$ is the total number of associated nodes to an element $e$, in which a material point $p$ resides.

The algorithm outlined in Fig. 4 (the function `[mpD] = SdS(meD,mpD,p2N)` called at the beginning of each cycle, see Fig. 4) represents the vectorised solution of the computation of basis functions and their derivatives.

Coordinates of the material points `mpD.x(:,1:2)` are first replicated and then subtracted by their associated nodes co-ordinates, e.g., `meD.x(p2N)` and `meD.y(p2N)` respectively (Lines 3 or 5 in Fig. 4). This yields the array `D` with the same dimension of `p2N`. This array of distance between the points and their associated nodes is sent as an input to the nested function `[N,dN] = NdN(D,h,lp)`, which computes 1D basis function and its derivative through matrix element-wise operations (operator `.*`) (either in Line 4 for $x$ coordinates or Line 6 for $y$ coordinates in Fig. 4).

Given the piece-wise Eq. 11, three logical arrays (`c1`, `c2` and `c3`) are defined (Lines 21-24 in Fig. 4), whose elements are either 1 (the condition is true) or 0 (the condition is false). Three arrays of basis functions are calculated (`N1`, `N2` and `N3`, Lines 26-28) according to Eq. 12. The array of basis functions `N` is obtained through a summation of the element-wise multiplications of these temporary arrays with their corresponding logical arrays (Line 29 in Fig. 4). The same holds true for the calculation of the gradient basis function (Lines 31-34 in Fig. 4). It is faster to use logical arrays as multipliers of precomputed basis function arrays rather than using these in a conditional indexing statement, e.g., `N(c2==1) = 1-abs(dX(c2==1))./h`. The performance gain is significant between the two approaches, i.e., an intrinsic 30 % gain over the wall-clock time of the

basis functions and derivatives calculation. We observe an invariance of such gain with respect to the initial number of material point per element or to the mesh resolution.

```matlab
1  function [mpD] = SdS(meD,mpD,p2N)
2  %% COMPUTE (X,Y)-BASIS FUNCTION
D       = (repmat(mpD.x(:,1),1,meD.nNe) - meD.x(p2N)) ;%
[Sx,dSx] = NdN(D,meD.h(1),repmat(mpD.l(:,1),1,meD.nNe));%
D       = (repmat(mpD.x(:,2),1,meD.nNe) - meD.y(p2N)) ;%
[Sy,dSy] = NdN(D,meD.h(2),repmat(mpD.l(:,2),1,meD.nNe));%
7  %% CONVOLUTION OF BASIS FUNCTIONS
mpD.S    =  Sx.* Sy                                    ;%
mpD.dSx  = dSx.* Sy                                    ;%
mpD.dSy  =  Sx.*dSy                                    ;%
11 %% B MATRIX ASSEMBLY
iDx            = 1:meD.DoF:meD.nDoF(1)-1               ;%
iDy            = iDx+1                                 ;%
mpD.B(1,iDx,:) = mpD.dSx'                              ;%
mpD.B(2,iDy,:) = mpD.dSy'                              ;%
mpD.B(3,iDx,:) = mpD.dSy'                              ;%
mpD.B(3,iDy,:) = mpD.dSx'                              ;%
end
function [N,dN]=NdN(dX,h,lp)
20 %% COMPUTE BASIS FUNCTIONS
lp = 2*lp                                              ;%
c1 = ( abs(dX)< (  0.5*lp)                        )   ;%
c2 = ((abs(dX)>=(  0.5*lp)) & (abs(dX)<(h-0.5*lp)))   ;%
c3 = ((abs(dX)>=(h-0.5*lp)) & (abs(dX)<(h+0.5*lp)))   ;%
25 % BASIS FUNCTION
N1 = 1-((4*dX.^2+lp.^2)./(4*h.*lp))                   ;%
N2 = 1-(abs(dX)./h)                                   ;%
N3 = ((h+0.5*lp-abs(dX)).^2)./(2*h.*lp)               ;%
N  = c1.*N1+c2.*N2+c3.*N3                             ;%
30 % BASIS FUNCTION GRADIENT
dN1= -((8*dX)./(4*h.*lp))                             ;%
dN2= sign(dX).*(-1/h)                                 ;%
dN3=-sign(dX).*(h+0.5*lp-abs(dX))./(h*lp)             ;%
dN = c1.*dN1+c2.*dN2+c3.*dN3                          ;%
end
```

**Figure 4.** Code Fragment 1 shows the vectorised solution to the calculation of the basis functions and their derivatives within SdS.m. Table B1 lists the variables used.

### 3.3.2 Integration of internal forces

Another computationally expensive operation for MATLAB$^{©}$ is the mapping (or accumulation) of the material point contributions to their associated nodes. It is performed by the function p2Nsolve.m in the workflow of the solver.

The standard calculations for the material point contributions to the lumped mass $m_n$, the momentum $\boldsymbol{p}_n$, the external $\boldsymbol{f}_n^e$ and internal $\boldsymbol{f}_n^i$ forces are given by:

$$m_n = \sum_{p \in n} S_n(\boldsymbol{x}_p) m_p, \tag{15}$$

$$\boldsymbol{p}_n = \sum_{p \in n} S_n(\boldsymbol{x}_p) m_p \boldsymbol{v}_p, \tag{16}$$

$$\boldsymbol{f}_n^e = \sum_{p \in n} S_n(\boldsymbol{x}_p) m_p \boldsymbol{b}_p, \tag{17}$$

$$\boldsymbol{f}_n^i = \sum_{p \in n} v_p \boldsymbol{B}^T(\boldsymbol{x}_p) \boldsymbol{\sigma}_p, \tag{18}$$

with $m_p$ the material point mass, $\boldsymbol{v}_p$ the material point velocity, $\boldsymbol{b}_p$ the body force applied to the material point and $\boldsymbol{\sigma}_p$ the material point Cauchy stress tensor in the Voigt notation.

Once the mapping phase is achieved, the equations of motions are explicitly solved forward in time on the mesh. Nodal accelerations $\boldsymbol{a}_n$ and velocities $\boldsymbol{v}_n$ are given by:

$$\boldsymbol{a}_n^{t+\Delta t} = m_n^{-1}(\boldsymbol{f}_n^e - \boldsymbol{f}_n^i), \tag{19}$$

$$\boldsymbol{v}_n^{t+\Delta t} = m_n^{-1}\boldsymbol{p}_n + \Delta t \boldsymbol{a}_n^{t+\Delta t}. \tag{20}$$

Finally, boundary conditions are applied to the nodes that belong to the boundaries.

The vectorised solution comes from the use of the built-in function `accumarray( )` of MATLAB© combined with `reshape( )` and `repmat( )`. The core of the vectorization is to use p2N as a vector (i.e., flattening the array p2N(:) results in a row vector) of subscripts with `accumarray`, which accumulates material point contributions (e.g., mass or momentum) that share the same node.

In the function p2Nsolve (Code Fragment 2 shown in Fig. 5), the first step is to initialize nodal vectors (mass, momentum, forces, etc.) to zero (Lines 4-5 in Fig. 5). Then, temporary vectors (`m`, `p`, `f` and `fi`) of material point contributions (namely, mass, momentum, and external and internal forces) are generated (Lines 10-17 in Fig. 5). The accumulation (nodal summation) is performed (Lines 19-26 in Fig. 5) using either the flattened `p2n(:)` or `l2g(:)` (e.g., the global indices of nodes) as the vector of subscripts. Note that for the accumulation of material point contributions of internal forces, a short for-loop iterates over the associated node (e.g., from 1 to `meD.nNe`) of every material point to accumulate their respective contributions.

To calculate the temporary vector of internal forces (`fi` at Lines 15-17 in Fig. 5), the first step consists of the matrix multiplication of the strain-displacement matrix `mpD.B` with the material point stress vector `mpD.s`. The vectorised solution is given by i) element-wise multiplications of `mpD.B` with a replication of the transposed stress vector `repmat(reshape(mpD.s,size(mpD.` whose result is then ii) summed by means of the built-in function `sum( )` along the columns and, finally multiplied by a replicated transpose of the material point volume vector, e.g., `repmat(mpD.V',meD.nDoF(1),1)`.

To illustrate the numerical efficiency of the vectorised multiplication between a matrix and a vector, we have developed an iterative and vectorised solution of $\boldsymbol{B}(\boldsymbol{x}_p)^T \boldsymbol{\sigma}_p$ with an increasing $n_p$ and considering single (4 bytes) and double (8 bytes)

```
 1  function [meD] = p2Nsolve(meD,mpD,g,dt,l2g,p2N,bc)
 2  %% INITIALIZATION
 3  % NODAL VECTOR INITIALIZATION
 4  meD.m(:) = 0.0 ; meD.mr(:) = 0.0 ; meD.f(:) = 0.0 ; meD.d(:) = 0.0      ;%
 5  meD.a(:) = 0.0 ;   meD.p(:) = 0.0 ; meD.v(:) = 0.0 ; meD.u(:) = 0.0      ;%
 6  %% CONTRIBUTION TO NODES
 7  % PREPROCESSING
 8  m = reshape( mpD.S.*repmat(mpD.m,1,meD.nNe)        ,mpD.n*meD.nNe    ,1)    ;%
 p = reshape([mpD.S.*repmat(mpD.p(:,1),1,meD.nNe);...
mpD.S.*repmat(mpD.p(:,2),1,meD.nNe)],mpD.n*meD.nDoF(1),1)      ;%
f = reshape([mpD.S.*0.0                          ;...
mpD.S.*repmat(mpD.m,1,meD.nNe).*-g ],mpD.n*meD.nDoF(1),1)      ;%
fi= squeeze(sum(mpD.B.*repmat(reshape(mpD.s,size(mpD.s,1),1,mpD.n)...
14                  ,1,meD.nDoF(1)),1)).*repmat(mpD.V',meD.nDoF(1),1)          ;%
15  % CONTRIBUTION FROM p TO N
meD.m = accumarray(p2N(:),m,[meD.nN      1])                             ;%
meD.p = accumarray(l2g(:),p,[meD.nDoF(2) 1])                             ;%
meD.f = accumarray(l2g(:),f,[meD.nDoF(2) 1])                             ;%
for n = 1:meD.nNe
20      l = [(meD.DoF*p2N(:,n)-1);(meD.DoF*p2N(:,n))]                        ;%
meD.f = meD.f - accumarray(l,[fi(n*meD.DoF-1,:)';...
fi(n*meD.DoF  ,:)'],[meD.nDoF(2) 1])            ;%
end                                                                      %
24  %% SOLVE EXPLICIT MOMENTUM BALANCE EQUATION
25  % UPDATE GLOBAL NODAL INFORMATIONS
26  iDx       = 1:meD.DoF:meD.nDoF(2)-1                                      ;%
27  iDy       = iDx+1                                                        ;%
28  % COMPUTE GLOBAL NODAL FORCE
meD.d(iDx) = sqrt(meD.f(iDx).^2+meD.f(iDy).^2)                           ;%
meD.d(iDy) = meD.d(iDx)                                                  ;%
meD.f      = meD.f - meD.vd*meD.d.*sign(meD.p)                           ;%
32  % UPDATE GLOBAL NODAL MOMENTUM
meD.p      = meD.p + dt*meD.f                                            ;%
34  % COMPUTE GLOBAL NODAL ACCELERATION AND VELOCITY
meD.mr     = reshape(repmat(meD.m',meD.DoF,1),meD.nDoF(2),1)             ;%
iD         = meD.mr==0                                                   ;%
meD.a      = meD.f./meD.mr                                               ;%
meD.v      = meD.p./meD.mr                                               ;%
meD.a(iD)  = 0.0                                                         ;%
meD.v(iD)  = 0.0                                                         ;%
41  % BOUNDARY CONDITIONS: FIX DIRICHLET BOUNDARY CONDITIONS
meD.a(bc.x(:,1))=bc.x(:,2)                                               ;%
meD.a(bc.y(:,1))=bc.y(:,2)                                               ;%
meD.v(bc.x(:,1))=bc.x(:,2)                                               ;%
meD.v(bc.y(:,1))=bc.y(:,2)                                               ;%
end
```

**Figure 5.** Code Fragment 2 shows the vectorised solution to the nodal projection of material point quantities (e.g., mass and momentum) within the local function `p2Nsolve.m`. The core of the vectorization process is the extensive use of the built-in function of MATLAB[©] `accumarray( )`, for which we detail the main features in the text. Table B1 lists the variables used.

arithmetic precision. The wall-clock time increases with $n_p$ with a sharp transition for the vectorised solution around $n_p \approx$ 1000, as showed in Fig 6a. The mathematical operation requires more memory than available in the L2 cache (1024 kB under the CPU architecture used), which inhibits cache reuse (Dabrowski et al., 2008). A peak performance of at least 1000 Mflops, showed in Fig. 6b, is achieved when $n_p = 1327$ or $n_p = 2654$ for simple or double arithmetic precision respectively, i.e., it corresponds exactly to 1024 kB for both precisions. Beyond, the performance dramatically drops to approximately the half of the peak value. This drop is even more severe for a double arithmetic precision.

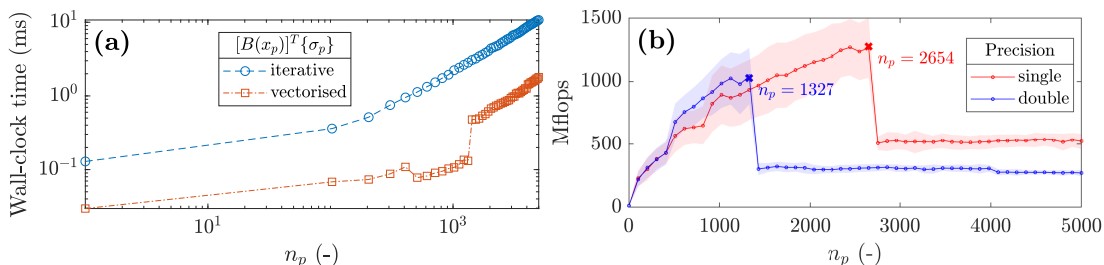

**Figure 6.** a) Wall-clock time to solve for a matrix multiplication between a multidimensional array and a vector with an increasing number of the third dimension with a double arithmetic precision and, b) number of floating point operations per second (flops) for single and double arithmetic precisions. The continuous line represents the averages value whereas the shaded area denotes the standard deviation.

### 3.3.3    Update of material point properties

Finally, we propose a vectorisation of the function `mapN2p.m` that i) interpolates updated nodal solutions to the material points (velocities and coordinates) and ii) the double mapping (DM or MUSL) procedure (see Fern et al. 2019). The material point velocity $\boldsymbol{v}_p$ is defined as an interpolation of the solution of the updated nodal accelerations, given by:

$$\boldsymbol{v}_p^{t+\Delta t} = \boldsymbol{v}_p^t + \Delta t \sum_{n=1}^{n_n} S_n(\boldsymbol{x}_p)\boldsymbol{a}_n^{t+\Delta t}. \tag{21}$$

The material point updated momentum is found by $\boldsymbol{p}_p^{t+\Delta t} = m_p \boldsymbol{v}_p^{t+\Delta t}$. The double mapping procedure of the nodal velocity $\boldsymbol{v}_n$ consists of the remapping of the updated material point momentum on the mesh, divided by the nodal mass, given as:

$$\boldsymbol{v}_n^{t+\Delta t} = m_n^{-1} \sum_{p \in n} S_n(\boldsymbol{x}_p)\boldsymbol{p}_p^{t+\Delta t}, \tag{22}$$

and for which boundary conditions are enforced. Finally, the material point coordinates are updated based on the following:

$$\boldsymbol{x}_p^{t+\Delta t} = \boldsymbol{x}_p^t + \Delta t \sum_{n=1}^{n_n} S_n(\boldsymbol{x}_p)\boldsymbol{v}_n^{t+\Delta t}. \tag{23}$$

To solve for the interpolation of updated nodal solutions to the material points, we rely on a combination of element-wise matrix multiplication between the array of basis functions `mpD.S` with the global vectors through a transform of the `p2N` array,

i.e., `iDx=meD.DoF*p2N-1` and `iDy=iDx+1` (Lines 3-4 in Code Fragment 3 in Fig. 7), which are used to access to x and y components of global vectors.

When accessing global nodal vectors by means of `iDx` and `iDy`, the resulting arrays are naturally of the same size as `p2N` and are therefore dimension-compatible with `mpD.S`. For instance, a summation along the columns (e.g., the associated nodes of material points) of an element-wise multiplication of `mpD.S` with `meD.a(iDx)` results in an interpolation of the x-component of the global acceleration vector to the material points.

This procedure is used for the velocity update (Line 6 in Fig. 7) and for the material point coordinate update (Line 22 in Fig. 7). A remapping of the nodal momentum is carried out (Lines 11 to 14 in Fig. 7), which allows calculating the updated nodal incremental displacements (Line 15 in Fig. 7). Finally, boundary conditions of nodal incremental displacements are enforced (Lines 19-20 in Fig. 7).

```
1  function [meD,mpD] = mapN2p(meD,mpD,dt,l2g,p2N,bc)
2  %% INTERPOLATE SOLUTIONS N to p
3  iDx        = meD.DoF*p2N-1                                    ;%
4  iDy        = iDx+1                                            ;%
5  % VELOCITY UPDATE
6  mpD.v      = mpD.v+dt*[sum(mpD.S.*meD.a(iDx),2) sum(mpD.S.*meD.a(iDy),2)] ;%
7  % MOMENTUM UPDATE
8  mpD.p      = mpD.v.*repmat(mpD.m,1,meD.DoF)                   ;%
9  %% UPDATE NODAL MOMENTUM WITH UPDATED MP MOMENTUM (MUSL OR DOUBLE MAPPING)
meD.p(:) = 0.0                                                ;%
p          = reshape([mpD.S.*repmat(mpD.p(:,1),1,meD.nNe) ;...
mpD.S.*repmat(mpD.p(:,2),1,meD.nNe)],...
mpD.n*meD.nDoF(1),1             )       ;%
meD.p    = accumarray(l2g(:),p,[meD.nDoF(2) 1])              ;%
meD.u    = dt*(meD.p./meD.mr)                                ;%
iD       = meD.mr==0                                         ;%
meD.u(iD)= 0.0                                               ;%
18 %% BOUNDARY CONDITIONS: FIX DIRICHLET BOUNDARY CONDITIONS
meD.u(bc.x(:,1))=bc.x(:,2)                                   ;%
meD.u(bc.y(:,1))=bc.y(:,2)                                   ;%
21 %% UPDATE COORDINATE AND DISPLACEMENT
mpD.x    = mpD.x+[sum(mpD.S.*meD.u(iDx),2) sum(mpD.S.*meD.u(iDy),2)]     ;%
mpD.u    = mpD.u+[sum(mpD.S.*meD.u(iDx),2) sum(mpD.S.*meD.u(iDy),2)]     ;%
end
```

**Figure 7.** Code Fragment 3 shows the vectorised solution for the interpolation of nodal solutions to material points with a double mapping procedure (or MUSL) within the function `mapN2p.m`.

**3.4 Initial settings and adaptive time step**

Regarding the initial setting of the background mesh of the demonstration cases further presented, we select a uniform mesh and a regular distribution of material points within the initially populated elements of the mesh. Each element is evenly filled with 4 material points, e.g., $n_{pe} = 2^2$, unless otherwise stated.

In this contribution, Dirichlet boundary conditions are resolved directly on the background mesh, as in the standard finite 320 element method. This implies that boundary conditions are resolved only in contiguous regions between the mesh and the material points. Deviating from this contiguity or having the mesh not aligned with the coordinate system requires specific treatments for boundary conditions (Cortis et al., 2018). Furthermore, we ignore the external tractions as their implementation is complex.

As explicit time integration is only conditionally stable, any explicit formulation requires a small time step $\Delta t$ to ensure numerical stability (Ni and Zhang, 2020), e.g., smaller than a critical value defined by the Courant-Friedrich-Lewy (CFL) condition. Hence, we employ an adaptive time step (de Vaucorbeil et al., 2020), which considers the velocity of the material points. The first step is to compute the maximum wave speed of the material using (Zhang et al., 2016; Anderson Jr, 1987)

$$(c_x, c_y) = \left( \max_p (V + | (v_x)_p |), \max_p (V + | (v_y)_p |) \right), \tag{24}$$

where the wave speed is $V = ((K + 4G/3)/\rho)^{\frac{1}{2}}$, $K$ and $G$ are the bulk and shear moduli respectively, $\rho$ is the material density, $(v_x)_p$ and $(v_y)_p$ are the material point velocity components. $\Delta t$ is then restricted by the CFL condition as followed:

$$\Delta t = \alpha \min \left( \frac{h_x}{c_x}, \frac{h_y}{c_y} \right), \tag{25}$$

where $\alpha \in [0; 1]$ is the time step multiplier, and $h_x$ and $h_y$ are the mesh spacings.

## 4 Results

In this section, we first demonstrate our MATLAB-based MPM solver to be efficient in reproducing results from other studies, i.e., the compaction of an elastic column (Coombs et al., 2020) (e.g., quasi-static analysis), the cantilever beam problem (Sadeghirad et al., 2011) (e.g., large elastic deformation) and an application to landslide dynamics (Huang et al., 2015) (e.g., elasto-plastic behaviour). Then, we present both the efficiency and the numerical performances for a selected case, e.g., the elasto-plastic collapse. We conclude and compare the performances of the solver with respect to the specific case of an impact of two elastic disks previously implemented in a Julia language environment by (Sinaie et al., 2017).

Regarding the performance analysis, we investigate the performance gain of the vectorised solver considering a double arithmetic precision with respect to the total number of material point because of the following reasons: i) the mesh resolution, i.e., the total number of elements $n_{el}$, influences the wall-clock time of the solver by reducing the time step due to the CFL condition hence increasing the total number of iterations. In addition, ii) the total number of material points $n_p$ increases the number of operations per cycle due to an increase of the size of matrices, i.e., the size of the strain-displacement matrix depends on $n_p$ and not on $n_{el}$. Hence, $n_p$ consistently influences the performance of the solver whereas $n_{el}$ determines the wall-clock time of the solver. The performance of the solver is addressed through both the number of floating point operations per second (flops), and by the average number of iteration per second (it/s). The number of floating point operations per second was manually estimated for each function of the solver.

### 4.1 Validation of the solver and numerical efficiency

#### 4.1.1 Convergence: elastic compaction under self-weight of a column

Following the convergence analysis proposed by Coombs and Augarde (2020); Wang et al. (2019); Charlton et al. (2017), we analyse an elastic column of an initial height $l_0 = 10$ m subjected to an external load (e.g. the gravity). We selected the

cpGIMPM variant with a domain update based on the diagonal components of the deformation gradient. Coombs et al. (2020) showed that such domain update is well suited for hydrostatic compression problems. We also selected the CPDI2q variant as a reference, because of its superior convergence accuracy for such problem compared to GIMPM (Coombs et al., 2020).

The initial geometry is shown in Fig. 8. The background mesh is made of a bi-linear four-noded quadrilaterals, and roller boundary conditions are applied on the base and the sides of the column, initially populated by 4 material points per element. The column is 1 element wide and $n$ elements tall and, the number of element in the vertical direction is increased from 1 to a maximum of 1280 elements. The time step is adaptive and we selected a time step multiplier of $\alpha = 0.5$, e.g., minimal and maximal time step values of $\Delta t_{\min} = 3.1 \cdot 10^{-4}$ s and $\Delta t_{\max} = 3.8 \cdot 10^{-4}$ s respectively for the finest mesh of 1280 elements.

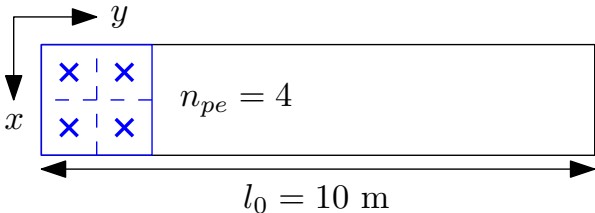

**Figure 8.** Initial geometry of the column.

To consistently apply the external load for the explicit solver, we follow the recommendation of Bardenhagen and Kober (2004), i.e., a quasi-static solution (given an explicit integration scheme is chosen) is obtained if the total simulation time is equal to 40 elastic wave transit times. The material has a Young's modulus $E = 1 \cdot 10^4$ Pa and a Poisson's ratio $\nu = 0$ with a density $\rho = 80$ kg m$^{-3}$. The gravity $g$ is increased from 0 to its final value, i.e., $g = 9.81$ m s$^{-2}$. We performed additional implicit quasi-static simulations (named iCPDI2q) in order to consistently discuss the results with respect to what was reported in Coombs and Augarde (2020). The external force is consistently applied over 50 equal load steps. The vertical normal stress is given by the analytical solution (Coombs and Augarde, 2020) $\sigma_{yy}(y_0) = \rho g(l_0 - y_0)$, where $l_0$ is the initial height of the column and $y_0$ is the initial position of a point within the column.

The error between the analytical and numerical solutions is as follows:

$$\text{error} = \sum_{p=1}^{n_p} \frac{||(\sigma_{yy})_p - \sigma_{yy}(y_p)|| (V_0)_p}{(\rho g l_0) V_0}, \tag{26}$$

where $(\sigma_{yy})_p$ is the stress along the $y$-axis of a material point $p$ (Fig. 8) of an initial volume $(V_0)_p$ and $V_0$ is the initial volume of the column, i.e., $V_0 = \sum_{p=1}^{n_p} (V_0)_p$.

The convergence toward a quasi-static solution is shown in Fig. 9 (a). Is is quadratic for both cpGIMPM and CPDI2q, but contrary to Coombs et al. (2020); Coombs and Augarde (2020) who reported a full convergence, it stops at error $\approx 2 \cdot 10^{-6}$ for the explicit implementation. This was already outlined by Bardenhagen and Kober (2004) as a saturation of the error caused by resolving the dynamic stress wave propagation, which is inherent to any explicit scheme. Hence, a static solution could never be achieved because, unlike quasi-static implicit methods, the elastic waves propagate indefinitely and the static equilibrium is never resolved. This is consistent when compared to the iCPDI2q solution we implemented, whose behaviour is still converging

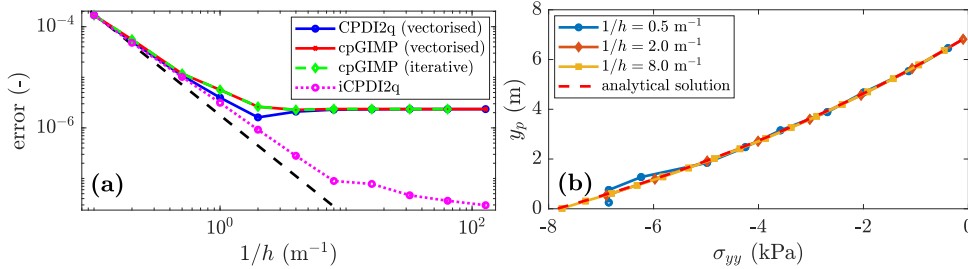

**Figure 9.** a) Convergence of the error: a limit is reached at error $\approx 2 \cdot 10^{-6}$ for the explicit solver, whereas the quasi-static solution still converges. This was already demonstrated in Bardenhagen and Kober (2004) as an error saturation due to the explicit scheme, i.e., the equilibrium is never resolved. b) The stress $\sigma_{yy}$ along the $y$-axis predicted at the deformed position $y_p$ by the CPDI2q variant is in good agreements with the analytical solution for a refined mesh.

below the limit error $\approx 2 \cdot 10^{-6}$ reached by the explicit solver. However, the convergence rate of the implicit algorithm decreases
as the mesh resolution increases. We did not investigate this since our focus is on the explicit implementation. The vertical stresses of material points are in good agreements with the analytical solution (see Fig. 9 b)). Some oscillations are observed for a coarse mesh resolution but these rapidly decrease as the mesh resolution increases.

We finally report the wall-clock time for the cpGIMPM (iterative), cpGIMPM (vectorised) and the CPDI2q (vectorised) variants. As claimed by Sadeghirad et al. (2013, 2011), the CPDI2q variant induces no significant computational cost compared
to the cpGIMPM variant. However, the absolute value between vectorised and iterative implementations is significant. For $n_p = 2560$, the vectorised solution completed in 1161 s whereas the iterative solution completed in 52'856 s. The vectorised implementation is roughly 50 times faster than the iterative implementation.

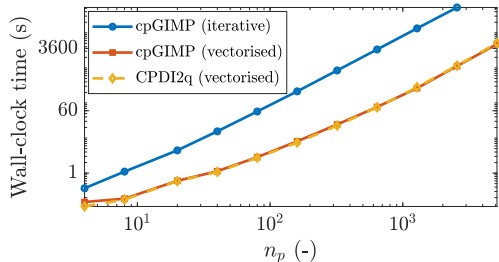

**Figure 10.** Wall-clock time for cpGIMPM (vectorised and iterative solutions) and the CPDI2q solution with respect to the total number of material points $n_p$. There is no significant differences between CPDI2q and cpGIMPM variants regarding the wall-clock time. The iterative implementation is also much slower than the vectorised implementation.

#### 4.1.2 Large deformation: the elastic cantilever beam problem

The cantilever beam problem (Sinaie et al., 2017; Sadeghirad et al., 2011) is the second benchmark which demonstrates the

390 robustness of the MPM solver. Two MPM variants are implemented, namely, i) the contiguous GIMPM (cpGIMPM) which relies on the stretch part of the deformation gradient (see Charlton et al. 2017) to update the particle domain since large rotations are expected during the deformation of the beam, and ii) the convected particle domain interpolation (CPDI, Leavy et al. 2019; Sadeghirad et al. 2011). We selected the CPDI variant since it is more suitable to large torsional deformation modes (Coombs et al., 2020) than the CPDI2q variant. Two constitutive elastic models are selected, i.e., neo-Hookean Guilkey and

395 Weiss (2003) or linear elastic York et al. (1999) solids. For consistency, we use the same physical quantities as in Sadeghirad et al. (2011), i.e., an elastic modulus $E = 10^6$ Pa, a Poisson's ratio $\nu = 0.3$, a density $\rho = 1050$ kg/m$^3$, the gravity $g = 10.0$ m/s and a real-time simulation $t = 3$ s with no damping forces introduced.

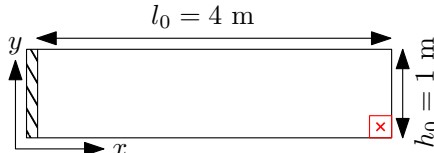

**Figure 11.** Initial geometry for the cantilever beam problem; the free end material point appears in red where a red cross marks its centre.

The beam geometry is depicted in Fig. 11 and is discretized by 64 four-noded quadrilaterals, each of them initially populated by 9 material points (e.g., $n_p = 576$) with a adaptive time step determined by the CFL condition, i.e., the time step multiplier is

400 $alpha = 0.1$, which yields minimal and maximal time step values of $\Delta t_{\min} = 5.7 \cdot 10^{-4}$ s and $\Delta t_{\max} = 6.9 \cdot 10^{-4}$ s respectively. The large deformation is initiated by suddenly applying the gravity at the beginning of the simulation, i.e., $t = 0$ s.

As indicated in Sadeghirad et al. (2011), the cpGIMPM simulation failed when using the diagonal components of the deformation gradient to update the material point domain, i.e., the domain vanishes under large rotations as stated in (Coombs et al., 2020). However and as as expected, the cpGIMPM simulation succeeded when using the diagonal terms of the stretch

part of the deformation gradient, as proposed by Coombs et al. (2020); Charlton et al. (2017). The numerical solutions, obtained by the latter cpGIMPM and CPDI, to the vertical deflection $\Delta u$ of the material point at the bottom free end of the beam (e.g., the red cross in Fig. 11) are shown in Fig. 12. Some comparative results reported by Sadeghirad et al. (2011) are depicted by black markers (squares for the FEM solution and circles for the CPDI solution), whereas the results of the solver are depicted by lines.

The local minimal and the minimal and maximal values (in timing and magnitude) are in agreement with the FEM solution of Sadeghirad et al. (2011). The elastic response is in agreement with the CPDI results reported by Sadeghirad et al. (2011) but, it differs in timing with respect to the FEM solution. This confirms our numerical implementation of CPDI when compared to the one proposed by Sadeghirad et al. (2011). In addition, the elastic response does not substantially differ from a linear elastic solid to a neo-Hookean one. It demonstrates the incremental implementation of the MPM solver to be relevant in capturing

large elastic deformations for the cantilever beam problem.

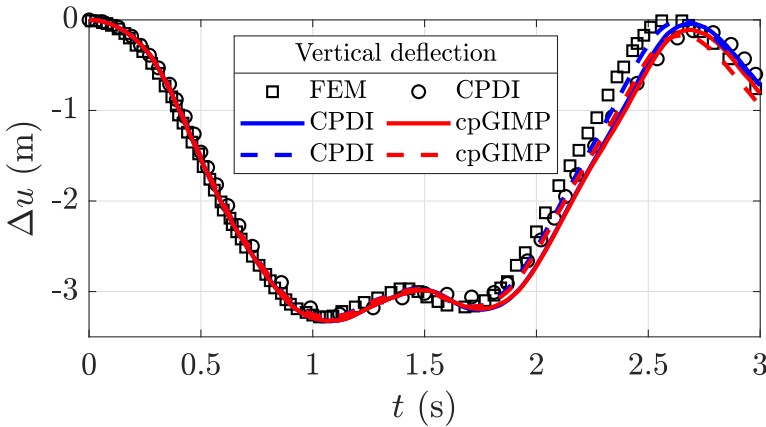

**Figure 12.** Vertical deflection $\Delta u$ for the cantilever beam problem. The black markers denote the solutions of Sadeghirad et al. (2011) (circles for CPDI and squares for FEM). The line colour indicates the MPM variant (blue for CPDI and red for cpGIMP), solid lines refer to a linear elastic solid, whereas dashed lines refer to a neo-Hookean solid. $\Delta u$ corresponds to the vertical displacement of the bottom material point at the free end of the beam (the red cross in Fig. 11).

Figure 13 shows the finite deformation of the material point domain, i.e., a) or c), and, the vertical Cauchy stress field, i.e., b) or d), for CPDI and cpGIMPM. The stress oscillations due to the cell-crossing error are partially cured when using a domain-based variant compared to the standard MPM. However, spurious vertical stresses are more developed in Fig. 13 (d) compared to Fig. 13 (b) where the vertical stress field appears even smoother. Both CPDI and cpGIMPM give a decent representation of the actual geometry of the deformed beam.

We also report a quite significant difference in execution time between the CPDI variant compared to the CPDI2q and cpGIMPM variants, i.e., CPDI executes in an average 280.54 it/s whereas both CPDI2q and CPGIMPM execute in an average 301.42 it/s and an average 299.33 it/s, respectively.

### 4.1.3 Application: the elasto-plastic slumping dynamics

We present an application of the MPM solver (vectorised and iterative version) to the case of landslide mechanics. We selected the domain-based CDPI variant since it performs better than the CPDI2q variant in modelling torsional and stretch deformation modes (Wang et al., 2019) coupled to an elasto-plastic constitutive model based on a non-associated Mohr-Coulomb (M-C) plasticity (Simpson, 2017). We i) analyse the geometrical features of the slump and, ii) compare the results (the geometry and the failure surface) to the numerical simulation of Huang et al. (2015), which is based on a Drucker-Prager model with tension cut-off (D-P).

The geometry of the problem is shown in Fig. 14, the soil material is discretized by $110 \times 35$ elements with $n_{pe} = 9$, resulting in $n_p = 21'840$ material points. A uniform mesh spacing $h_{x,y} = 1$ m is used and, rollers are imposed at the left and right domain limits while a no-slip condition is enforced at the base of the material. We closely follow the numerical procedure proposed in

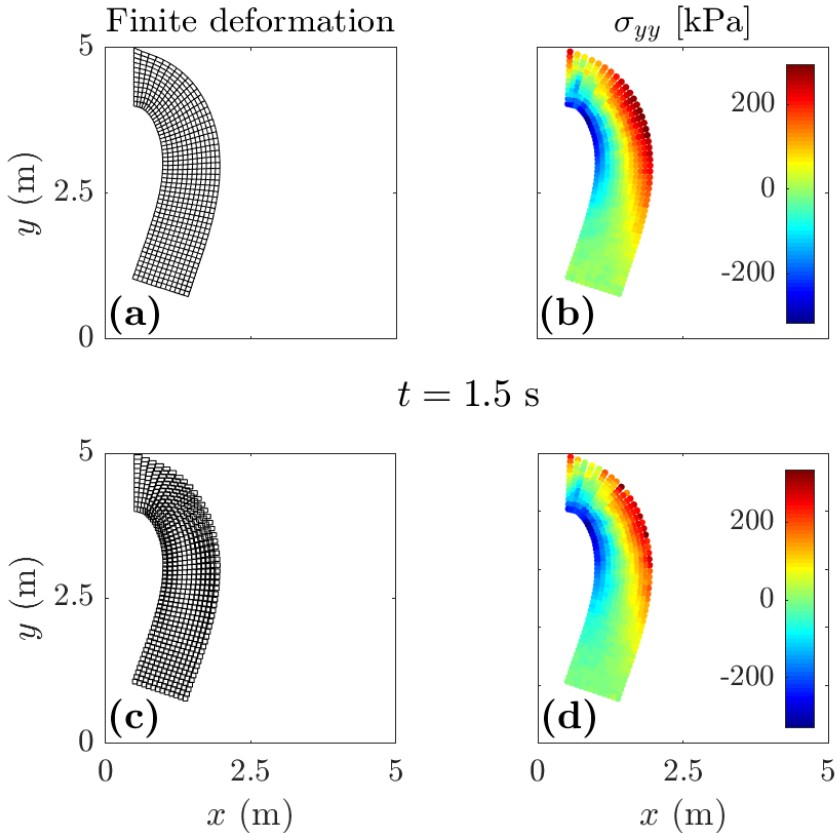

**Figure 13.** Finite deformation of the material point domain and vertical Cauchy stress $\sigma_{yy}$ for CPDI, i.e., a) & b), and for cpGIMPM, i.e., c) & d). The CPDI variant gives a better and contiguous description of the material point's domain and a slightly smoother stress field, compared to the cpGIMPM variant, which is based on the stretch part of the deformation gradient.

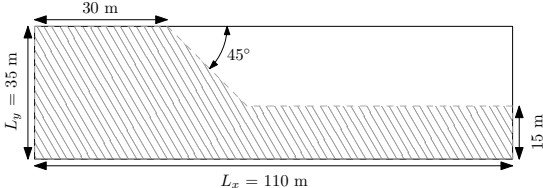

**Figure 14.** Initial geometry for the slump problem from Huang et al. (2015). Roller boundary conditions are imposed on the left and right of the domain while a no-slip condition is enforced at the base of the material.

Huang et al. (2015), i.e., no local damping is introduced in the equation of motion and the gravity is suddenly applied at the beginning of the simulation. As in Huang et al. (2015), we consider an elasto-plastic cohesive material of density $\rho = 2100$ kg·m$^3$, with an elastic modulus $E = 70$ MPa and a Poisson's ratio $\nu = 0.3$. The cohesion is $c = 10$ Pa, the internal friction angle is $\phi = 20°$ with no dilatancy, i.e., the dilatancy angle is $\psi = 0$. The total simulation time is 7.22 s and, we select a time step multiplier $\alpha = 0.5$. The adaptive time steps (considering the elastic properties and the mesh spacings $h_{x,y} = 1$ m) yield minimal and maximal values $\Delta t_{\min} = 2.3 \cdot 10^{-3}$ s and $\Delta t_{\max} = 2.4 \cdot 10^{-3}$ s respectively.

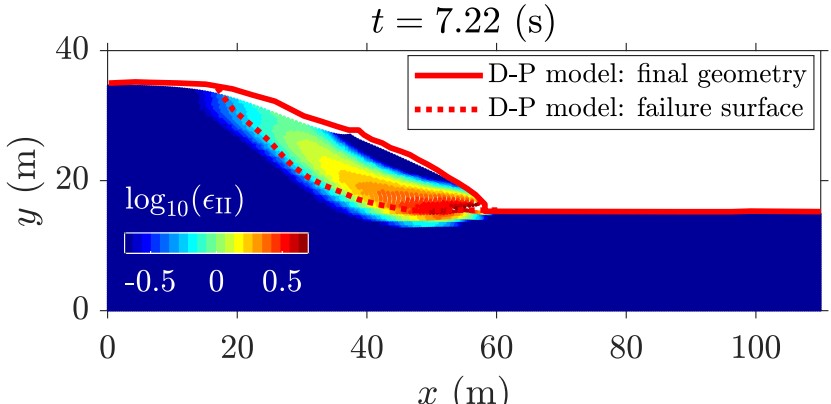

**Figure 15.** MPM solution to the elasto-plastic slump. The red lines indicate the numerical solution of Huang et al. (2015) and, the coloured points indicate the second invariant of the accumulated plastic strain $\epsilon_{II}$ obtained by the CPDI solver. An intense shear zone progressively develops backwards from the toe of the slope, resulting in a circular failure mode.

The numerical solution to the elasto-plastic problem is shown in Fig. 15. An intense shear zone, highlighted by the second invariant of the accumulated plastic strain $\epsilon_{II}$, develops at the toe of the slope as soon as the material yields and propagates backwards to the top of the material. It results in a rotational slump. The failure surface is in good agreement with the solution reported by Huang et al. (2015) (continuous and discontinuous red lines in Fig. 15) but, we also observe differences, i.e., the crest of the slope is lower compared to the original work of Huang et al. (2015). This may be explained by the problem of spurious material separation when using sMPM or GIMPM (Sadeghirad et al., 2011), the latter being overcome with the CPDI variant, i.e., the crest of the slope experiences considerable stretch deformation modes. Despite some differences, our numerical results appear coherent with those reported by Huang et al. (2015).

The vectorised and iterative solutions are resolved within approximately 630 s (a wall-clock time of $\approx$ 10 min. and an average 4.20 it/s) and 14'868 s (a wall-clock time of $\approx$ 4.1 hrs. and an average 0.21 it/s) respectively. This corresponds to a performance gain of 23.6. The performance gain is significant between an iterative and a vectorised solver for this problem.

## 4.2 Computational performance

### 4.2.1 Iterative and vectorised elasto-plastic collapses

We evaluate the computational performance of the solver, using the MATLAB version R2018a on an Intel Core i7-4790, with a benchmark based on the elasto-plastic collapse of the aluminium-bar assemblage, for which numerical and experimental results were initially reported by Bui et al. (2008) and Huang et al. (2015) respectively.

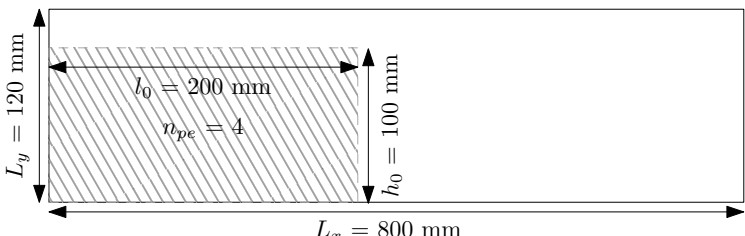

**Figure 16.** Initial geometry for the elasto-plastic collapse (Huang et al., 2015). Roller boundaries are imposed on the left and right boundaries of the computational domain while a no-slip condition is enforced at the bottom of the domain. The aluminium-bar assemblage has dimensions of $l_0 \times h_0$ and is discretized by $n_{pe} = 4$ material points per initially populated element.

We vary the number of elements of the background mesh, which results in a variety of different regular mesh spacings $h_{x,y}$. The number of elements along the x- and y- directions are $n_{el,x} = [10, 20, 40, 80, 160, 320, 640]$ and $n_{el,y} = [1, 2, 5, 11, 23, 47, 95]$ respectively. The number of material points per element is kept constant, i.e., $n_{pe} = 4$, and this yields a total number of material points $n_p = [10, 50, 200, 800, 3'200, 12'800, 51'200]$. The initial geometry and boundary conditions used for this problem are depicted in Fig. 16. The total simulation time is 1.0 s and, the time step multiplier is $\alpha = 0.5$. Accordingly to Huang et al. (2015), the gravity $g = 9.81$ m·s$^{-2}$ is applied to the assemblage and, no damping is introduced. We consider a non-cohesive granular material (Huang et al., 2015) of density $\rho = 2650$ kg·m$^3$, with a bulk modulus $K = 0.7$ MPa and a Poisson's ratio $\nu = 0.3$. The cohesion is $c = 0$ Pa, the internal friction angle is $\phi = 19.8°$ and there is no dilatancy, i.e., $\psi = 0$.

We conducted preliminary investigations using either uGIMPM or cpGIMPM variants, the latter with a domain update based either on the determinant of the deformation gradient or on the diagonal components of the stretch part of the deformation gradient. We concluded the uGIMPM was the most reliable, even though its suitability is restricted to both simple shear and pure rotation deformation modes (Coombs et al., 2020).

We observe a good agreement between the numerical simulation and the experiments (see Fig. 17), considering either the final surface (blue square dotted line) or the failure surface (blue circle dotted line). The repose angle in the numerical simulation is approximately $13°$, which is in agreements with the experimental data reported by Bui et al. (2008), e.g., they reported a final angle of $14°$.

The vectorised and iterative solutions (for a total of $n_p = 12'800$ material points) are resolved within approximately 1595 s (a wall-clock time of $\approx 0.5$ hrs. and an average 10.98 it/s) and 43'861 s (a wall-clock time of $\approx 12$ hrs. and an average 0.38

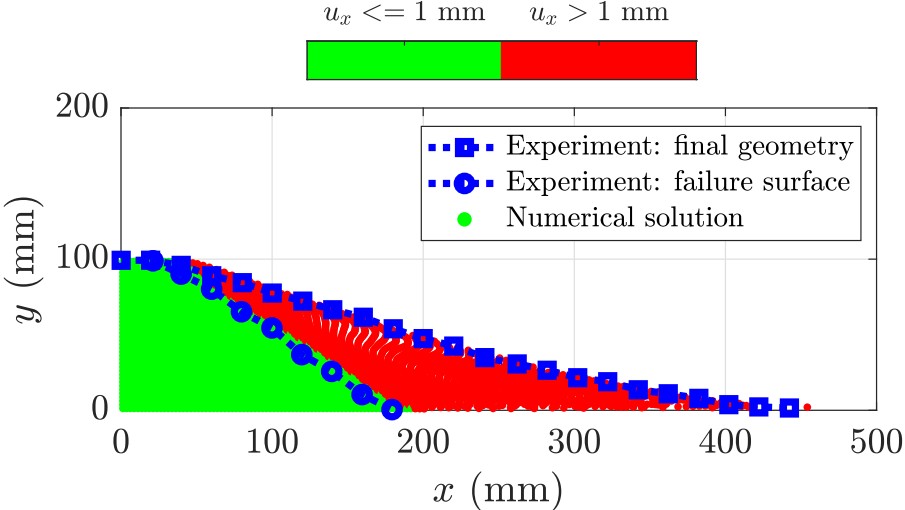

**Figure 17.** Final geometry of the collapse: in the intact region (horizontal displacement $u_x < 1$ mm), the material points are coloured in green, whereas in the deformed region (horizontal displacement $u_x > 1$ mm), they are coloured in red and indicate plastic deformations of the initial mass. The transition between the deformed and undeformed region marks the failure surface of the material. Experimental results of (Bui et al., 2008) are depicted by the blue dotted lines. The computational domain is discretized by a background mesh made of 320 $\times$ 48 quadrilateral elements with $n_p = 4$ per initially populated element, i.e., a total $n_p = 12'800$ material points discretize the aluminium assemblage.

it/s) respectively. This corresponds to a performance gain of 28.24 for a vectorised code over an iterative code to solve this
elasto-plastic problem.

The performance of the solver is demonstrated in Fig. 18. A peak performance of $\approx 900$ Mflops is reached, as soon as $n_p$ exceeds 1000 material points and, a residual performance of $\approx 600$ Mflops is further resolved (for $n_p \approx 50'000$ material points). Every functions provide an even and fair contribution on the overall performance, except the function `constitutive.m` for which the performance appears delayed or shifted. First of all, this function treats the elasto-plastic constitutive relation, in
which the dimensions of the matrices are smaller when compared to the other functions. Hence, the amount of floating point operations per second is lower compared to other functions, e.g., `p2Nsolve.m`. This results in less performance for an equivalent number of material points. It also requires a greater number of material points to increase the dimensions of the matrices in order to exceed the L2 cache maximum capacity.

This considerations provide a better understanding of the performance gain of the vectorised solver showed in Fig. 19: the
gain increases and then, reaches a plateau and ultimately, decreases to a residual gain. This is directly related to the peak and the residual performances of the solver showed in Fig. 18.

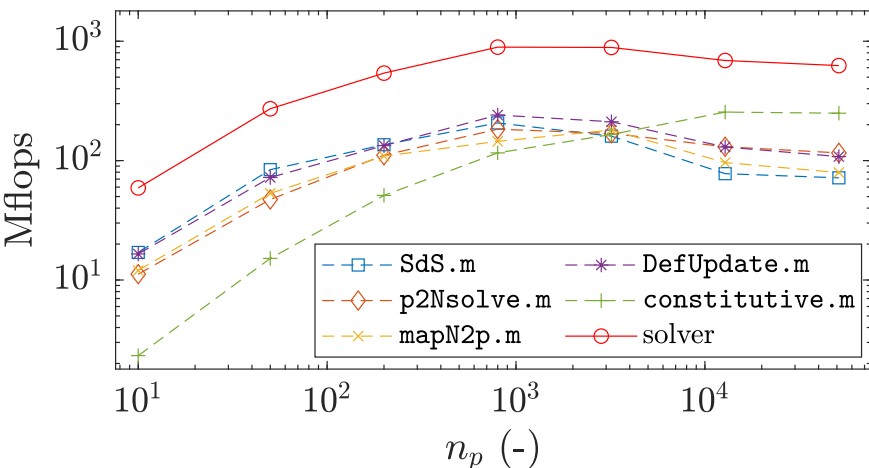

**Figure 18.** Number of floating point operation per seconds (flops) with respect to the total number of material point $n_p$ for the vectorised implementation. The discontinuous lines refer to the functions of the solver, whereas the continuous line refer to the solver. A peak performance of 900 Mflops is reached by the solver for $n_p > 1000$ and, a residual performance of 600 Mflops is further resolved for an increasing $n_p$.

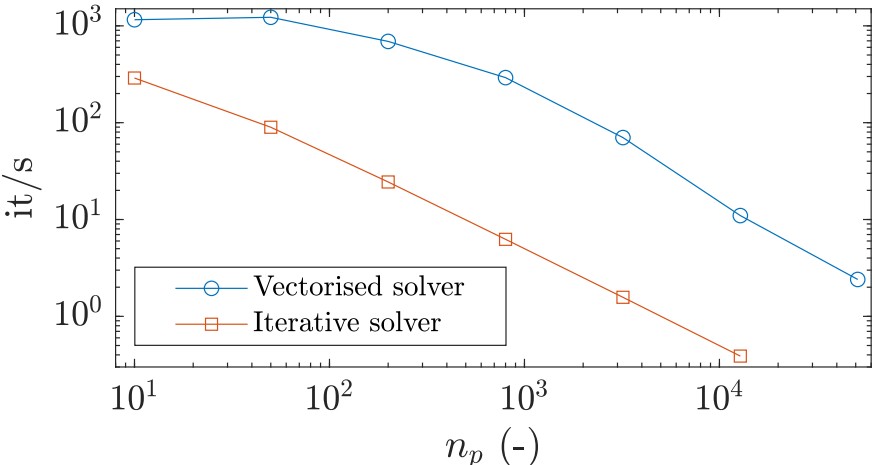

**Figure 19.** Number of iterations per second with respect to the total number of material point $n_p$. The greatest performance gain is reached around $n_p = 1000$, which is related to the peak performance of the solver (see Fig. 18). The gains corresponding to the peak and residual performances are 46 and 28 respectively.

#### 4.2.2 Comparison between Julia and MATLAB

We compare the computational efficiency of the vectorised CPDI2q MATLAB implementation and the computational efficiency reported by Sinaie et al. (2017) of a Julia-based implementation of the collision of two elastic disks problem. However, we note a difference between the actual implementation and the one used by Sinaie et al. (2017); the latter is based on a USL variant with a cut-off algorithm, whereas the present implementation relies on the MUSL (or double mapping) procedure, which necessitates a double mapping procedure. The initial geometry and parameters are the same as those used in Sinaie et al. (2017). However, the time step is adaptive and, we select a time step multiplier $\alpha = 0.5$. Given the variety of mesh resolution, we do not present minimal and maximal time step values.

Our CPDI2q implementation, in MATLAB R2018a, is, at least, 2.8 times faster than the Julia implementation proposed by Sinaie et al. (2017) for similar hardware (see Table 1). Sinaie et al. (2017) completed the analysis with an Intel Core i7-6700 (4 cores with a base frequency of 3.40 GHz up to a turbo frequency of 4.00 GHz) with 16 GB RAM, whereas we used an Intel Core i7-4790 with similar specifications (see Section 2). However, the performance ratio between MATLAB and Julia seems to decrease as the mesh resolution increases.

**Table 1.** Efficiency comparison of the Julia implementation of Sinaie et al. (2017), and the MATLAB-based implementation for the two elastic disk impact problems.

| mesh | $n_{pe}$ | $n_p$ | Its/s | | |
|------|----------|-------|-------|--------|------|
| | | | Julia | MATLAB | Gain |
| $20 \times 20$ | $2^2$ | 416 | 132.80 | 450.27 | 3.40 |
| $20 \times 20$ | $4^2$ | 1'624 | 33.37 | 118.45 | 3.54 |
| $40 \times 40$ | $2^2$ | 1'624 | 26.45 | 115.59 | 4.37 |
| $80 \times 80$ | $4^2$ | 25'784 | 1.82 | 5.21 | 2.86 |

## 5 Discussion

In this contribution, a fast and efficient explicit MPM solver is proposed that considers two variants (e.g., the uGIMPM/cpGIMPM and the CPDI/CPDI2q variants).

Regarding the compression of the elastic column, we report a good agreement of the numerical solver with previous explicit MPM implementations, such as Bardenhagen and Kober (2004). The same flaw of an explicit scheme is also experienced by the solver, i.e., a saturation of the error due to the specific usage of an explicit scheme that resolves the wave propagation, thus preventing any static equilibrium to be reached. This confirms that our implementation is consistent with previous MPM implementations. However, the implicit implementation suffers from a decrease of the convergence rate for a fine mesh resolution. Further work would be needed to investigate this decrease of convergence rate. This case also demonstrated that cpGIMPM

and CPDI variants have a similar computational cost and, this confirms the suitability of cpGIMPM with respect to CPDI, as previously mentioned by Coombs et al. (2020); Charlton et al. (2017).

For the cantilever beam, we report a good agreement of the solver with the results of Sadeghirad et al. (2011), i.e., we report the vertical deflection of the beam to be very close in both magnitude and timing (for the CPDI variant) to the FEM solution. However, we also report a slower execution time for the CPDI variant when compared to both cpGIMPM and CPDI2q variants.

The elasto-plastic slump also demonstrates the solver to be efficient in capturing complex dynamics in the field of geomechanics. The CDPI solution showed that the algorithm proposed by Simpson (2017) to return stresses when the material yields is well suited to the slumping dynamics. However and as mentioned by Simpson (2017), such return mapping is only valid under the assumption of a non-associated plasticity with no volumetric plastic strain. This particular case of isochoric plastic deformations rises the issue of volumetric locking. In the actual implementation, no regularization techniques are considered. As a result, the pressure field experiences severe locking for isochoric plastic deformations. One way to overcome locking phenomenons would be to implement the regularization technique initially proposed by Coombs et al. (2018) for quasi-static sMPM and GIMPM implementations.

Regarding the elasto-plastic collapse, the numerical results demonstrate the solver to be in agreement with both previous experimental and numerical results (Huang et al., 2015; Bui et al., 2008). This confirms the ability of the solver to address elasto-plastic problems. However, the choice of whether to update or not the material point domain remains critical. Such question remains open and would require a more thorough investigation of the suitability of each of these domain updating variants. Nevertheless, the uGIMPM variant is a good candidate since, i) it is able to reproduce the experimental results of Bui et al. (2008) and, ii) it ensures numerical stability. However, one has to keep in mind its limited range of suitability regarding the deformation modes involved. If a cpGIMPM is selected, the splitting algorithm proposed in Gracia et al. (2019); Homel et al. (2016) could be implemented to mitigate the amount of distortion experienced by the material point domains during deformation. We did not selected the domain updating method based on the corners of the domain as suggested in Coombs et al. (2020). This is because such domain updating method necessitates to calculate additional shape functions between the corners of the domain of the material point with their associated nodes. This results in an additional computational cost. Nevertheless, such variant is of interest and should be addressed as well when the computational performances are not the main concern.

The computational performance comes from the combined use of the connectivity array `p2N` with the built-in function `accumarray( )` to i) accumulate material point contributions to their associated nodes or, ii) to interpolate the updated nodal solutions to the associated material points. When a residual performance is resolved, an overall performance gain (e.g., the amount of it/s) of 28 is reported. As an example, the functions `p2nsolve.m` and `mapN2p.m` are 24 and 22 times faster than an iterative algorithm when the residual performance is achieved. The overall performance gain is in agreement to other vectorised FEM codes, i.e., O'Sullivan et al. (2019) reported an overall gain of 25.7 for a optimised continuous Galerkin finite element code.

An iterative implementation would require multiple nested for-loops and a larger number of operations on smaller matrices, which increase the number of BLAS calls, thus inducing significant BLAS overheads and decreasing the overall performance

of the solver. This is limited by a vectorised code structure. However and as showed by the matrix multiplication problem, the L2 cache reuse is the limiting factor and, it ultimately affects the peak performance of the solver due to these numerous RAM-to-cache communications for larger matrices. Such problem is serious and, its influence is demonstrated by the delayed response in terms of performance for the function `constitutive.m`. However, we also have to mention that the overall residual performance was resolved only for a limited total number of material points. The performance drop of the function `constitutive.m` has never been achieved. Consequently, we suspect an additional decrease of overall performances of the solver for larger problems.

The overall performance achieved by the solver is higher than expected and, is even higher with respect to what was reported by Sinaie et al. (2017). We demonstrate that MATLAB is even more efficient than Julia, i.e., a minimum 2.86 performance gain achieved compared to a similar Julia CPDI2q implementation. This confirms the efficiency of MATLAB for solid mechanics problems, provided a reasonable amount of time is spent on the vectorisation of the algorithm.

## 6  Conclusions

We have demonstrated the capability of MATLAB as an efficient language in regard to a material point method (MPM) implementation in an explicit formulation when bottleneck operations (e.g., calculations of the shape function or material point contributions) are properly vectorised. The computational performances of MATLAB are even higher than those previously reported for a similar CPDI2q implementation in Julia, provided that built-in functions such as `accumarray( )` are used. However, the numerical efficiency naturally decreases with the level of complexity of the chosen MPM variant (sMPM, GIMPM or CPDI/CPDI2q).

The vectorisation activities we performed provide a fast and efficient MATLAB-based MPM solver. Such vectorised code could be transposed to a more efficient language, such as the C-CUDA language, that is known to efficiently take advantage of vectorised operations.

As a final word, a future implementation of a poro-elasto-plastic mechanical solver could be applied to complex geomechanical problems such as landslide dynamics while benefiting from a faster numerical implementation in C-CUDA, thus resolving high three-dimensional resolutions in a decent and affordable amount a time.

*Code availability.* The fMPMM-solver developed in this study is licensed under the GPLv3 free software licence. The latest version of the code is available for download from Bitbucket at: https://bitbucket.org/ewyser/fmpmm-solver/src/master/ (last access: October 6, 2020). The fMPMM-solver archive (v1.0 and v1.1) is available from a permanent DOI repository (Zenodo) at: https://doi.org/10.5281/zenodo.4068585 (Wyser et al., 2020). The fMPMM-solver software includes the reproducible codes used for this study.

## Appendix A: Acronyms

**Table A1.** Acronyms used throughout the manuscript

| | |
|---|---|
| PIC | **P**article-**i**n-**C**ell |
| FLIP | **FL**uid **I**mplicit **P**article |
| FEM | **F**inite **E**lement **M**ethod |
| sMPM | **s**tandard **M**aterial **P**oint **M**ethod |
| GIMPM | **G**eneralized **M**aterial **P**oint **M**ethod |
| uGIMPM | **u**ndeformed **G**eneralized **M**aterial **P**oint **M**ethod |
| cpGIMPM | **c**ontiguous **p**article **G**eneralized **M**aterial **P**oint **M**ethod |
| CPDI | **C**onvected **P**article **D**omain **I**nterpolation |
| CPDI2q | **C**onvected **P**article **D**omain **I**nterpolation **2**nd order **q**uadrilateral |

## Appendix B: fMPMM-solver Variables

**Table B1.** Variables of the structure arrays for the mesh `meD` and the material point `mpD` used in Code Fragment 1 & 2 shown in Figs. 4 & 5. `nDF` stores the local and global number of degrees of freedom, i.e., `nDF=[nNe,nN*DoF]`. The constant `nstr` is the number of stress components, according to the standard definition of the Cauchy stress tensor using the Voigt notation, e.g., $\boldsymbol{\sigma}_p = (\sigma_{xx}, \sigma_{yy}, \sigma_{xy})$.

|  | Variable | Description | Dimension |
|---|---|---|---|
|  | nNe | nodes per element | (1) |
|  | nN | number of nodes | (1) |
|  | DoF | degree of freedom | (1) |
|  | nDF | number of DoF | (1,2) |
|  | h | mesh spacing | (1,DoF) |
| meD. | x | node coordinates | (nN,1) |
|  | y | node coordinates | (nN,1) |
|  | m | nodal mass | (nN,1) |
|  | p | nodal momentum | (nDF(2),1) |
|  | f | nodal force | (nDF(2),1) |
|  | n | number of points | (1) |
|  | l | domain half-length | (np,DoF) |
|  | V | volume | (np,1) |
|  | m | mass | (np,1) |
|  | x | point coordinates | (np,DoF) |
| mpD. | p | momentum | (np,DoF) |
|  | s | stress | (np,nstr) |
|  | S | basis function | (np,nNe) |
|  | dSx | derivative in x | (np,nNe) |
|  | dSy | derivative in y | (np,nNe) |
|  | B | B matrix | (nstr,nDF(1),np) |

*Author contributions.* EW wrote the original manuscript and developed, together with YP, the first version the solver (fMPMM-solver, v1.0). YA provided technical supports, assisted EW in the revision of the latest version of the solver (v1.1) and corrected specific parts of the solver. EW and YA wrote together the revised version of the manuscript. MJ and YP supervised the early stages of the study and provided guidance. All authors have reviewed and approved the final version of the paper.

*Competing interests.* The authors declare that they have no conflicts of interest.

*Acknowledgements.* Yury Alkhimenkov gratefully acknowledges the support from the Swiss National Science Foundation (grant no. 172691). Yury Alkhimenkov and Yury Y. Podladchikov gratefully acknowledge support from the Russian Ministry of Science and Higher Education (project No. 075-15-2019-1890). The authors gratefully thank Johan Gaume for his comments that contributed to improve the overall quality of the manuscript.

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
