# Peer review of "Fast and efficient MATLAB-based MPM solver (fMPMM-solver v1.1)"

_Geoscientific Model Development, 2020_

## Referee Comment (RC1) · Anonymous Referee #1 · 11 Aug 2020

The paper presents an explicit vectorised form of the material point method for the generalised interpolation and CPDI2 variants of the method. The paper is reasonably written but is missing many details on the implemented algorithms and the numerical analyses are too focused on reproducing the results of others rather than on numerical performance which is the main trust of the article.

I have the following specific comments:

- the introduction to the paper appears to have picked a random selection of MPM articles rather than focusing on articles that look at the numerical implementation of the method. The introduction should be made more coherent and focused.

- the authors have not picked up on any of the papers that extend the MILAMIN ap-

proach. In particular the authors should refer to the following paper that extends the MILAMIN ideas to non-linear problems: O'Sullivan, S, Bird, R.E., Coombs, W.M. & Giani, S. (2019). Rapid non-linear finite element analysis of continuous and discontinuous Galerkin methods in MATLAB. Computers and Mathematics with Applications 78(9): 3007-3026. There are others and the authors should review the appropriate literature.

- If performance is the focus, why use MATLAB? Why not adopt one of the existing MPM codes that are written in compiled code which will always be faster than MATLAB? Such as: jixiefx.com, github.com/yuanming-hu/taichi\_mpm, cimne.com/kratos/, github.com/nairnj/nairn-mpm-fea, github.com/cb-geo/mpm/, sourceforge.net/p/mpmgimp, github.com/xzhang66/MPM3D-F90. There may be others.

- the authors mention an implicit implementation but do not present any results in terms of the speed gains of the implicit vectorised algorithm. These should be included and the vectorised algorithm explained.

- CPDI2 methods suffer from issues associated with domain distortion and are not suitable for problems involving large shear/rotation. This point should be acknowledged, see for example: Wang, L., Coombs, W.M. , Augarde, C.E. , Cortis, M. Charlton, T.J. , Brown, M.J. Knappett, J., Brennan, A. Davidson, C. Richards, D. & Blake, A. (2019). On the use of domain-based material point methods for problems involving large distortion. Computer Methods in Applied Mechanics and Engineering 355: 1003-1025.

- What large deformation formulation has been used in this paper in terms of stresses and strains? Logarithmic strains and Kirchhoff stresses? Explain and justify the large deformation framework.

- How has the plasticity algorithm been implemented within the large deformation framework? The authors mention a prediction/correction type algorithm so how have they recovered the additive decomposition of the elastic and plastic strains from the

multiplicative decomposition of the deformation gradient? Critical details are missing here.

- How are boundary conditions imposed in this model?

- Figure 6 - what causes the step in the red line around 10ˆ3 material points? Is there a shift in terms of the cost within the algorithm?

- It is well known that MPMs can provide a reasonable global approximation (in terms of force-displacement response) but the computed stress field can be spurious/highly oscillatory due to issues such as locking and/or cell crossing. The authors should present the predicted stress response for the numerical examples presented in the paper and compare them to analytical solutions where available.

- The elastic column problem will only have around 2% deformation in terms of the deformed to original height so this problem is not a good test of the convergence of MPMs. The authors refer to the work of Coombs et al. for this problem but they show convergence for the case where the column compresses to around 50% of its initial height. Run the convergence for the case with a Young's modulus of 10kPa, not 1MPa.

- How have the authors avoided volumetric locking when using isochoric plastic flow? What do the pressure distributions look like through the deformed domains?

- the various comments on the use of cpGIMP and uGIMP are confused and misleading. Just because an analysis is stable it does not mean that it is "appropriate" as the results may be meaningless. The authors cite the work of Coombs to justify the use of the stretch to update the domains, however this technique does not work for problems involving simple shear type deformation, refer to Table 2 of: Coombs, WM, Augarde, CE, Brennan, AJ, Brown, MJ, Charlton, TJ, Knappett, JA, Ghaffari Motlagh, Y & Wang, L (2020). On Lagrangian mechanics and the implicit material point method for large deformation elasto-plasticity. Computer Methods in Applied Mechanics and Engineering 358: 112622. The authors later adopt a determinant of the deformation

gradient-type updating method but this has been shown to not converge for simple compression problems as the domains artificially shrink in the non-compressed direction and overlap in the compressed direction (see Figure 8 from the above reference). For example, the statement on lines 76-77 is not correct, or rather it is only correct for some types of problem.

- The speed gains of MILAMIN come from the combination of blocking and vectorisation. Have both techniques been used in this paper? If so, what are the relative speed gains from the different sources? How does the speed gain change with different block sizes?

- The authors have not explained by MATLAB is inefficient when working with small amounts of data. This point should be discussed with reference to CPU cache size and RAM-to-cache overheads.

- It would be more appropriate to present the speed gains in terms of flops. There should be a peak in performance if you consider large enough problems.

Other minor points: - Figure 2 is misleading as the GIMP domain is fully inside a single element and will have the same connectivity as the standard MP case. - Line 162 - the authors mention a 30% speed up - what problem/size of problem/number of points, etc? - where are material points located within the elements when the problems are set up? - some key information is missing from the numerical analysis, such as timestep sizes and total times which would make it impossible to reproduce the results. This information must be added. - incomplete sentence on line 359.

---

## Author Comment (AC1) · 14 Aug 2020

First, we would like to thank the referee for the time spent on reviewing our paper and for his stimulating comments.

**Referee's comment 1**

The paper presents an explicit vectorised form of the material point method for the generalised interpolation and CPDI2 variants of the method. The paper is reasonably written but is missing many details on the implemented algorithms and the numerical analyses are too focused on reproducing the results of others rather than on numerical performance which is the main trust of the article.

**Author's reply 1**

[Figure]

Thank you for the time spent on reviewing our work. We acknowledge the lack of details on the implemented algorithm. We will develop the numerical implementation in the revised manuscript. In addition, it is true that the numerical analyses are more focused on reproducing results of other studies. Therefore, we will investigate further the numerical performance of the solver in the revised manuscript, i.e., a detail analysis of performance will be presented for a wide range of numerical setup (increasing mesh resolutions, initial number of material points per element, etc.). This should provide a more exhaustive analysis of the numerical performance.

**Referee's comment 2**

The introduction to the paper appears to have picked a random selection of MPM articles rather than focusing on articles that look at the numerical implementation of the method. The introduction should be made more coherent and focused.

**Author's reply 2**

We acknowledge the lack of coherence of the introduction section. Consequently, we will focus our introduction toward the numerical implementation of MPM and add references which focus on numerical implementation of MPM and FEM, since both share common grounds.

**Referee's comment 3**

The authors have not picked up on any of the papers that extend the MILAMIN approach. In particular the authors should refer to the following paper that extends the MILAMIN ideas to non-linear problems: O'Sullivan, S, Bird, R.E., Coombs, W.M. & Giani, S. (2019). Rapid non-linear finite element analysis of continuous and discontinuous Galerkin methods in MATLAB. Computers and Mathematics with Applications 78(9): 3007-3026. There are others and the authors should review the appropriate literature.

**Author's reply 3**

Indeed, we were not aware of further extension to the MILAMIN approach. We reviewed the reference proposed and found common approach concerning vectorization, i.e., the use of the accumarray function to accumulate internal force contributions to nodes. Additionally, we will review the appropriate literature and consequently update the introduction section.

**Referee's comment 4**

If performance is the focus, why use MATLAB? Why not adopt one of the existing MPM codes that are written in compiled code which will always be faster than MATLAB? Such as: jixiefx.com, github.com/yuanminghu/taichi\_mpm, cimne.com/kratos/, github.com/nairnj/nairn-mpm-fea, github.com/cbgeo/mpm/, source-forge.net/p/mpmgimp, github.com/xzhang66/MPM3D-F90. There may be others.

**Author's reply 4**

It is true that MPM codes written in compiled codes will always be faster than MATLAB. Our point is to show that one can use MATLAB to produce prototyped code with a decent computational performance. However, our true concern is to propose an efficient and vectorised MATLAB code, which could be easily translated into the C CUDA language, or, at least, which partially minimizes the hurdles of the syntax translation while preserving the algorithm. As such, we will clearly state such concern in our revised manuscript.

**Referee's comment 5**

The authors mention an implicit implementation but do not present any results in terms of the speed gains of the implicit vectorised algorithm. These should be included and the vectorised algorithm explained.

**Author's reply 5**

It is true that i) we do not present any results related to speed gain and, ii) the implicit implementation is not explained. Our concern was to show that the error saturation

we reported for the explicit implementation was due to the explicit formulation. Consequently, an implicit implementation under the vectorisation framework of the explicit formulation should converge without any saturation error. We showed it did. However, we can include in the Supplementary Material of the paper a description of the vectorization for the implicit implementation if needed.

**Referee's comment 6**

CPDI2 methods suffer from issues associated with domain distortion and are not suitable for problems involving large shear/rotation. This point should be acknowledged, see for example: Wang, L., Coombs, W.M. , Augarde, C.E. , Cortis, M. Charlton, T.J. , Brown, M.J. Knappett, J., Brennan, A. Davidson, C. Richards, D. & Blake, A. (2019). On the use of domain-based material point methods for problems involving large distortion. Computer Methods in Applied Mechanics and Engineering 355: 1003-1025.

**Author's reply 6**

It is true that CPDI2q suffers from domain distortion, as shown in the cited reference by the referee. We will clarify this in the revised manuscript. In general, we will discuss in greater details advantages and flaws of domain-based MPM.

**Referee's comment 7**

What large deformation formulation has been used in this paper in terms of stresses and strains? Logarithmic strains and Kirchhoff stresses? Explain and justify the large deformation framework.

**Author's reply 7**

The large deformation formulation is based on the infinitesimal strain coupled to an objective measure of the stress for the finite deformation, i.e., the Jaumann stress rate formulation. It is a widely accepted deformation framework in a number a study, see Bandara, S., Ferrari, A., Laloui, L. Modelling landslides in unsaturated slopes subjected to rainfall infiltration using material point method. International Journal for Numerical

and Analytical Methods in Geomechanics. 40, 1358-1380 and many others. We think for an explicit implementation with small time step, such deformation formulation stands to reason. These considerations will be specified in the revised paper and the large deformation framework we selected will be clearly stated.

**Referee's comment 8**

How has the plasticity algorithm been implemented within the large deformation framework? The authors mention a prediction/correction type algorithm so how have they recovered the additive decomposition of the elastic and plastic strains from the multiplicative decomposition of the deformation gradient? Critical details are missing here.

**Author's reply 8**

It is true that critical details are missing concerning the plasticity algorithm we selected. At first, we did not want to emphasize return algorithm used in this study. But since the concern of the referee, we will add more details concerning the way plasticity is handle in the revised manuscript. More specifically, the prediction/correction algorithm used in this study combined with the small strain theory and the objective stress measure allow us to afford the decomposition of the elastic and plastic strain from the decomposition of the deformation gradient, as done in the MATLAB code AMPLE. We preferred the approach of Huang, P., Li, S-L., Guo, H., Hao, Z-M. Large deformation failure analysis of the soil slope based on the material point method. Computational Geosciences. 19. 951-963.

**Referee's comment 9**

How are boundary conditions imposed in this model?

**Author's reply 9**

Boundary conditions can be difficult to impose thoroughly in MPM, especially when considering non-aligned boundary conditions with respect to the background mesh, as mentioned in Cortis, M., Coombs, W., Augarde, C., Brown, M., Brennan, A., Robinson,

S. Imposition of essential boundary conditions in the material point method. International Journal for Numerical Methods in Engineering. 113: 130-152. This is the main reason why we choose numerical problem and setup in which boundary conditions could be directly applied on the background mesh. We will mention this important point of aligned boundary conditions in the revised manuscript.

**Referee's comment 10**

Figure 6 - what causes the step in the red line around 10Ë Ę3 material points? Is there a shift in terms of the cost within the algorithm?

**Author's reply 10**

Thank you for this comment. We started an investigation in terms of FLOPS and L2 cache concerning this step around 10ˆ3 material points and we just concluded this shift is due to the overhead and RAM-to-cache transfer. As mentioned further by the referee, overheads are a major concern in MATLAB. This step typically occurs when the block of data send from RAM to the L2 cache exceeds its maximum capacity. MATLAB can longer treat information as a contiguous block of data sent one time and start swapping from RAM to cache and back and forth. This costs an additional computational expense which results in this step in Figure 6 in the paper (see FIG. I).

FIG I: a) Floating-point operation per second in million and, b) linear increase of RAM usage with an increasing number of material point. The limit when the allocated space in RAM reaches the L2 cache size (1024kB) corresponds to the drop of FLOPS in FIG I a). Thanks to the referee's comment, we now come with a different approach to vectorise the matrix multiplication problem which is faster than the previous one but still suffers from overhead due RAM-to-cache communication. For this particular problem, we report a rather significant difference between the two vectorization options. This is particularly obvious when we observe the difference in terms of MFLOPS: FLOPS are at least twice higher than the former implementation for the matrix multiplication (see FIG. II).

FIG II: a) Floating-point operation per second in million for old and new vectorisation and, b) wall-clock time in ms for an increasing number of material point for an iterative calculation and the old and new vectorisation.

Consequently, we will replace the previous vectorisation by this new technique and make the necessary changes in our revised paper. It also implies to modify results and observations in the computational efficiency section of the paper. However, we will also mention the drop of efficiency due to overheads. In addition, this cost due to RAM-to-cache communication would be an interesting improvement of a future implementation of a vectorised MPM code in MATLAB.

**Referee's comment 11**

It is well known that MPMs can provide a reasonable global approximation (in terms of force-displacement response) but the computed stress field can be spurious/highly oscillatory due to issues such as locking and/or cell crossing. The authors should present the predicted stress response for the numerical examples presented in the paper and compare them to analytical solutions where available.

**Author's reply 11**

This is a good suggestion. We will present the stress field for numerical problems in the revised manuscript. Additionally, we will also present the analytical solution of the vertical stress for the elastic column problem and its analytical solution.

**Referee's comment 12**

The elastic column problem will only have around 2% deformation in terms of the deformed to original height so this problem is not a good test of the convergence of MPMs. The authors refer to the work of Coombs et al. for this problem but they show convergence for the case where the column compresses to around 50% of its initial height. Run the convergence for the case with a Young's modulus of 10kPa, not 1MPa.

**Author's reply 12**

[Figure]

The referee is totally right; the elastic column will only have a small vertical deformation. We will run the convergence analysis with lower elastic modulus as proposed to achieve a greater deformation of the column.

**Referee's comment 13**

How have the authors avoided volumetric locking when using isochoric plastic flow? What do the pressure distributions look like through the deformed domains?

**Author's reply 13**

We did not avoid volumetric locking when using isochoric plastic flow. It is true that any MPM variant suffers from volumetric locking. We should mention this particular problem when treating elasto-plastic problem and we will also add figure to visualize the pressure field for related problems. Additional discussions of flaws due to volumetric locking will be elaborated in the discussion section.

**Referee's comment 14**

The various comments on the use of cpGIMP and uGIMP are confused and misleading. Just because an analysis is stable it does not mean that it is "appropriate" as the results may be meaningless. The authors cite the work of Coombs to justify the use of the stretch to update the domains, however this technique does not work for problems involving simple shear type deformation, refer to Table 2 of: Coombs, WM, Augarde, CE, Brennan, AJ, Brown, MJ, Charlton, TJ, Knappett, JA, Ghaffari Motlagh, Y & Wang, L (2020). On Lagrangian mechanics and the implicit material point method for large deformation elasto-plasticity. Computer Methods in Applied Mechanics and Engineering 358: 112622. The authors later adopt a determinant of the deformation gradient-type updating method but this has been shown to not converge for simple compression problems as the domains artificially shrink in the non-compressed direction and overlap in the compressed direction (see Figure 8 from the above reference). For example, the statement on lines 76-77 is not correct, or rather it is only correct for

some types of problem.

**Author's reply 14**

We agree that this various comment might be confused and misleading. As such, we will elaborate the presentation of domain-based MPM and correctly refer to advantages and flaws for each of these domain-based variants. Regarding the determinant of the deformation gradient-type update, the referee comment is absolutely true. However, we selected such approach since it gave the most stable numerical results. But, we will clearly mention the problem of artificial shrink and overlap associated with this variant in the revised paper.

**Referee's comment 15**

The speed gains of MILAMIN come from the combination of blocking and vectorisation. Have both techniques been used in this paper? If so, what are the relative speed gains from the different sources? How does the speed gain change with different block sizes?

**Author's reply 15**

We did use vectorisation but we did not use blocking. It is true that in MILAMIN the speed gain come from a combination between vectorisation and blocking. However, such technique is applied in assembling local stiffness matrix in the global stiffness matrix, which does not appear in an explicit MPM formulation. Hence, the technique of blocking was disregarder for that concern.

**Referee's comment 16**

The authors have not explained by MATLAB is inefficient when working with small amounts of data. This point should be discussed with reference to CPU cache size and RAM-to-cache overheads.

**Author's reply 16**

We will discuss this point explicitly when presenting the new vectorisation framework

of the matrix multiplication problem we mentioned previously in our response.

**Referee's comment 17**

It would be more appropriate to present the speed gains in terms of flops. There should be a peak in performance if you consider large enough problems.

**Author's reply 17**

Thank you for this relevant suggestion. As mentioned previously in our reply, we already started to investigate the speed gain of the matrix-vector operation problem mentioned in Line 198. However, and as mentioned by the referee, we will extend this scope to the computational efficiency section which will greatly benefit from this.

**Referee's comment 18**

Figure 2 is misleading as the GIMP domain is fully inside a single element and will have the same connectivity as the standard MP case.

**Author's reply 18**

We agree Figure 2 is misleading. Consequently, we will replace the previous Figure 2 by a new one in the revised manuscript (see FIG. III), which we show below

FIG III: Proposed new Figure 2 which shows a correct representation of the connectivity for sMPM and GIMP.

**Referee's comment 19**

Line 162 - the authors mention a 30% speed up - what problem/size of problem/number of points, etc?

**Author's reply 19**

We mention a 30 % speed up at Line 162 for the calculation of shape function. It is true such statement lacks of evidence and we will provide the reader with a more thoroughly motivated argument. As suggested, we will analyse the performance gain for a variety

of number of material points per element and different mesh sizes considering the two kind of vectorization. Thank you for pointing that out.

**Referee's comment 20**

Where are material points located within the elements when the problems are set up?

**Author's reply 20**

Regarding the initial location of material points within elements, we preferred a regular distribution of material point within elements. Considering 4 material points per element, their location (in local coordinate) would be [-0.5; 0.5] along x and y direction. We will specify this in the revised paper.

**Referee's comment 21**

Some key information is missing from the numerical analysis, such as time step sizes and total times which would make it impossible to reproduce the results. This information must be added.

**Author's reply 21**

We also add information such as time step sizes and total times for the sake of further comparisons, as suggested.

**Referee's comment 22**

Incomplete sentence on line 359.

**Author's reply 22**

That is correct, sentence on line 359 will be completed in th
* * *
[Figure]

[Figure]

**Fig. 1.**

[Figure]

**Fig. 2.**

[Figure]

**Fig. 3.**

[Figure]

**Fig. 4.**

a) sMPM

3   6   9

7   11

2   8

$p = 1$

6   10

1   4   7

p2e $= \{5\}$

p2N $= \{6, 7, 10, 11\}$

b) GIMPM

3   6   9

3   7   11

2   8

6   10

$p = 1$

1   5   9

p2e $= \{1, 2, 4, 5\}$

p2N $= \{1, 2, 3, 5, 6, 7, 9, 10, 11\}$

**Fig. 5.**

---

## Referee Comment (RC2) · Anonymous Referee #2 · 24 Aug 2020

This manuscript presents a vectorized material point method for MATLAB and quantifies the increase in computational efficiency gained from using vectorized rather than iterative code. To my knowledge, this paper provides the only formal analysis of the performance gains from vectorizing MPM code. The presented vectorization approach could be easily implemented within existing and future MPM models. However, as already thoroughly noted by Referee #1, many details of the algorithms, setup of simulations, and numerical analysis are missing. I would like to add the following comments to those already given.

General comments:

-I found the structure of the paper to be confusing, especially in Section 4, where results from five test cases are reported. These test cases are all nearly exact reproductions

of previously-published work. The first two test cases – the elastic compaction of a column (Section 4.1) and elastic cantilever beam (Section 4.2) – appear to solely serve as benchmark examples for verification of the MPM model, though this is not clearly stated. The third test case – elasto-plastic column collapse (Section 4.3) – also appears to serve as further verification of the model until it is used again in Section 4.4, where the main results of the paper concerning the computational efficiency gained from vectorization are presented. A fourth test case (collision of two elastic disks) is also presented in 4.4 for further analysis of computational efficiency. The final test case (elasto-plastic landslide) is then presented in Section 4.5, which seems to serve the dual purpose of further model verification and a geomechanical application.

-The motivation behind most of the test cases is not clear, especially on the first read. Section 4 would benefit from a short introduction (before 4.1) that outlines what test cases were selected and for what purpose.

-It may help readability if the test cases for elasto-plastic column collapse and collision of two elastic disks are separated into an entirely different section from the rest of the examples, as these two test cases provide the main results in the paper regarding the computational efficiency gained by vectorization.

-Many of the statements regarding the effectiveness of cpGIMPM vs. uGIMPM vs. CPDI are misleading or lacking in detail:

-Line 288: in what way did domain updates based on the deformation gradient result in failure?

-Line 301: "[The elasto-plastic MPM solver] demonstrates the inability of the MPM variants based on a domain update (GIMPM or CPDI) to resolve extremely large plastic deformations when relying on the normal components of the deformation gradient or its stretch part to update the material point domain". This is too general of a statement. There are many cases in which these domain updates would work well; for example, if simple shear is minimal and the "stretch" update is used (Coombs et al 2020). A

similarly-flawed statement is made in the conclusion section (lines 394-396). Better conclusions regarding GIMPM/CPDI might incorporate the performance gains reported using the vectorization scheme for calculation of the shape functions and the difference in computational efficiency measured for GIMPM vs CPDI.

-Line 305: As already pointed out by Referee #1, it should be noted that the determinant of the deformation gradient-type GIMPM domain update is problematic for simple compression problems. Have the authors tried updating the GIMPM domains with the "corner" scheme from Eqs 35-37 in Coombs et al (2020)? Perhaps it would be more robust.

-There does not appear to be any reference to Fig. 1 in the main text. The caption for Fig. 1 also appears to lack any description of panels B and C.

-The list of the three steps of a typical MPM cycle at the beginning of Section 3.1 seems misplaced and is somewhat repetitive of the description of MPM in the previous sections. I suspect this list should link with Fig. 1 and may be more appropriately located within Section 2.

Minor comments:

- Fig 2. In the caption, "GIMP" should be changed to "GIMPM" to match the rest of the paper. The authors have already fixed the error in Fig. 2b regarding which nodes are associated with the GIMPM domain

-Line 39: which numerical considerations from MILAMIN are used? This is not specifically addressed

-several citations are missing parentheses (e.g. lines 52 and 84).

References:

Coombs, WM, Augarde, CE, Brennan, AJ, Brown, MJ, Charlton, TJ, Knappett, JA, Ghaffari Motlagh, Y & Wang, L (2020). On Lagrangian mechanics and the implicit

material point method for large deformation elasto-plasticity. Computer Methods in Applied Mechanics and Engineering 358: 112622.

---

## Author Comment (AC2) · 1 Sep 2020

**Referee's comment 1**

This manuscript presents a vectorized material point method for MATLAB and quantifies the increase in computational efficiency gained from using vectorized rather than iterative code. To my knowledge, this paper provides the only formal analysis of the performance gains from vectorizing MPM code. The presented vectorization approach could be easily implemented within existing and future MPM models. However, as already thoroughly noted by Referee #1, many details of the algorithms, setup of simulations, and numerical analysis are missing. I would like to add the following comments to those already given.

**Author's reply 1**

We agree many details of the algorithms are missing and, we will provide more details in the revised version, i.e. a clearer description of the constitutive model chosen, the large deformation framework based on the Jaumann rate. In addition, we will also provide further details concerning the setup of the simulation of the cases presented.

General comments:

**Referee's comment 2**

I found the structure of the paper to be confusing, especially in Section 4, where results from five test cases are reported. These test cases are all nearly exact reproductions of previously-published work. The first two test cases – the elastic compaction of a column (Section 4.1) and elastic cantilever beam (Section 4.2) – appear to solely serve as benchmark examples for verification of the MPM model, though this is not clearly stated. The third test case – elasto-plastic column collapse (Section 4.3) – also appears to serve as further verification of the model until it is used again in Section 4.4, where the main results of the paper concerning the computational efficiency gained from vectorization are presented. A fourth test case (collision of two elastic disks) is also presented in 4.4 for further analysis of computational efficiency. The final test case (elasto-plastic landslide) is then presented in Section 4.5, which seems to serve the dual purpose of further model verification and a geomechanical application.

**Author's reply 2**

It is true that the structure might be confusing. We will focus on Section 4 and make our point clearer by adding, as suggested in the following by the Referee, a quick introduction to this section. We think it would improve the readability of this section. We will also clearly state our motivation of exact reproduction of previously published work. We think it is also important to showcase the efficiency of our solver with benchmarks from past studies. However, we will reorganise the subsection in which computational

**[GMDD](https://example.com)**
performances are shown, in order to make our point more obvious to the reader.

**Referee's comment 3**

The motivation behind most of the test cases is not clear, especially on the first read. Section 4 would benefit from a short introduction (before 4.1) that outlines what test cases were selected and for what purpose.

**Author's reply 3**

Thank you for this comment. We agree that the motivation behind most of the test cases is not clear and, we will add a short introduction in Section 4, before 4.1 as suggested. We will also present the test cases and, we will motivate our choices for each of these to clarify our motivation.

**Referee's comment 4**

It may help readability if the test cases for elasto-plastic column collapse and collision of two elastic disks are separated into an entirely different section from the rest of the examples, as these two test cases provide the main results in the paper regarding the computational efficiency gained by vectorization.

**Author's reply 4**

It is a good suggestion and it should even significantly help the readability of the manuscript. We will create a new section focused on the computational efficiency, divided in two subsections, i.e., the first one for the elastic disk case and the second one for the elasto-plastic column collapse.

**Referee's comment 5**

Many of the statements regarding the effectiveness of cpGIMPM vs. uGIMPM vs. CPDI are misleading or lacking in detail:

**Author's reply 5**

We agree with the Referee that the statements related to the effectiveness of uGIMPM, cpGIMPM and CPDI are misleading or lacking in detail. Therefore, we created a new subsection 2.2 Domain-based material point method variants, which summarises the major difference between GIMPM and CPDI (the way the material point's domain is updated). We also present the different domain updating methods for GIMPM and we more clearly highlight the suitability of each of these domains updating methods based on the investigation of Coombs, WM, Augarde, CE, Brennan, AJ, Brown, MJ, Charlton, TJ, Knappett, JA, Ghaffari Motlagh, Y & Wang, L (2020). On Lagrangian mechanics and the implicit material point method for large deformation elasto-plasticity. Computer Methods in Applied Mechanics and Engineering 358: 112622, and, Wang, L., Coombs, W.M. , Augarde, C.E. , Cortis, M. Charlton, T.J. , Brown, M.J. Knappett, J., Brennan, A. Davidson, C. Richards, D. & Blake, A. (2019). On the use of domain-based material point methods for problems involving large distortion. Computer Methods in Applied Mechanics and Engineering 355: 1003-1025. Finally, we explicitly state at the end of new 2.2 subsection that the domain-based method as well as the domain update method should be carefully chosen accordingly to the deformation mode expected for a given case. Moreover, we also clearly state that the domain update method will be clearly stated in each case presented in the Results section. We believe that now this additional 2.2 subsection provides a better overview of both domain-based method and domain update method.

**Referee's comment 6**

Line 288: in what way did domain updates based on the deformation gradient result in failure? The domain updates based on the deformation gradient showed some flaws of GIMP-based implementation, i.e. the domain artificially shrinks under large rotation, which was already demonstrated in Fig.4 in Coombs, WM, Augarde, CE, Brennan, AJ, Brown, MJ, Charlton, TJ, Knappett, JA, Ghaffari Motlagh, Y & Wang, L (2020). On Lagrangian mechanics and the implicit material point method for large deformation elasto-plasticity. Computer Methods in Applied Mechanics and Engineering 358:

112622. As such, we will clearly mention the reason why the simulation in which the domain updates based on the deformation gradient failed.

**Referee's comment 7**

Line 301: "[The elasto-plastic MPM solver] demonstrates the inability of the MPM variants based on a domain update (GIMPM or CPDI) to resolve extremely large plastic deformations when relying on the normal components of the deformation gradient or its stretch part to update the material point domain". This is too general of a statement. There are many cases in which these domain updates would work well; for example, if simple shear is minimal and the "stretch" update is used (Coombs et al 2020). A similarly-flawed statement is made in the conclusion section (lines 394-396). Better conclusions regarding GIMPM/CPDI might incorporate the performance gains reported using the vectorization scheme for calculation of the shape functions and the difference in computational efficiency measured for GIMPM vs CPDI.

**Author's reply 7**

We agree that the statement on Line 301 is too general. As such, we will specify our concern regarding the deformation mode involved avoiding such flawed statements in the revised version of the manuscript, referring explicitly to the work of Coombs, WM, Augarde, CE, Brennan, AJ, Brown, MJ, Charlton, TJ, Knappett, JA, Ghaffari Motlagh, Y & Wang, L (2020). On Lagrangian mechanics and the implicit material point method for large deformation elasto-plasticity. Computer Methods in Applied Mechanics and Engineering 358: 112622. As suggested by the Referee, we will also focus our discussion on the performance gain of GIMPM and CPDI variants, which is indeed the main point of our contribution.

**Referee's comment 8**

Line 305: As already pointed out by Referee #1, it should be noted that the determinant of the deformation gradient-type GIMPM domain update is problematic for simple

compression problems. Have the authors tried updating the GIMPM domains with the "corner" scheme from Eqs 35-37 in Coombs et al (2020)? Perhaps it would be more robust.

**Author's reply 8**

This is a good comment. First, we will add a presentation of the suitability of domain updating methods in GIMPM for the possible deformation modes, directly referring to the work of Coombs, WM, Augarde, CE, Brennan, AJ, Brown, MJ, Charlton, TJ, Knappett, JA, Ghaffari Motlagh, Y & Wang, L (2020). On Lagrangian mechanics and the implicit material point method for large deformation elasto-plasticity. Computer Methods in Applied Mechanics and Engineering 358: 112622. This is important and is lacking up to now. Concerning the investigation of the corner update, we did not have tried to implement it. We decided to quickly investigate this scheme but we quickly figured out it was necessary to compute additional shape functions between material point's corners and nodes, similarly to the CPDI variant. Our concern comes from the need to compute these additional quantities, alongside to the shape function relating material points to their associated nodes. This extra cost is significant, especially when our major concern is computational performance. We therefore decided not to pursue such investigation for that reason.

**Referee's comment 9**

There does not appear to be any reference to Fig. 1 in the main text. The caption for Fig. 1 also appears to lack any description of panels B and C.

**Author's reply 9**

Indeed, there was no reference to Fig. 1 in the main text. We added a description of panels B and C as highlighted by the Referee and, we refer now to this Fig. 1 in the main text, as mentioned in the following reply #10.

**Referee's comment 10**

The list of the three steps of a typical MPM cycle at the beginning of Section 3.1 seems misplaced and is somewhat repetitive of the description of MPM in the previous sections. I suspect this list should link with Fig. 1 and may be more appropriately located within Section 2

**Author's reply 10**

We agree with the Referee; this list is misplaced. Consequently, we moved it in Section 2 and, we made a direct reference to this list with Fig. 1. Minor comments:

**Referee's comment 11**

Fig 2. In the caption, "GIMP" should be changed to "GIMPM" to match the rest of the paper. The authors have already fixed the error in Fig. 2b regarding which nodes are associated with the GIMPM domain

**Author's reply 11**

Right, we will make the change in the revised paper.

**Referee's comment 12**

Line 39: which numerical considerations from MILAMIN are used? This is not specifically addressed

**Author's reply 12**

Due to the comment of the first reviewer concerning the lack of focus in the introduction section, we partially have rewritten it. We now specify in the introduction section the considerations (from MILAMIN but also from the works of Birds et al, 2017 and O'Sullivan et al, 2019), which are the use of the function accumarray(), to reduce the number of BLAS calls.

**Referee's comment 13**

several citations are missing parentheses (e.g. lines 52 and 84).

**Author's reply 13**

We will fix these numerous citation problems in the revised paper.

---

## Author Response (AR1)

**Author responses to the Interactive discussion on "Fast an efficient MATLAB-based MPM solver (fMPMM-solver v1.0)" in the Geoscientific Model Development (GMD) Journal**

Emmanuel Wyser, Yury Alkhimenkov, Michel Jaboyedoff, Yury Podladchikov

October 12, 2020

The referee comments appear in black, whereas our responses appear in blue and the changes made in the revised manuscript appear in red.

**1 Referee #1**

The paper presents an explicit vectorised form of the material point method for the generalised interpolation and CPDI2 variants of the method. The paper is reasonably written but is missing many details on the implemented algorithms and the numerical analyses are too focused on reproducing the results of others rather than on numerical performance which is the main trust of the article.

Thank you for the time spent on the revision of our work. We acknowledge the lack of details on the implemented algorithm. We developed the numerical implementation, especially the deformation framework chosen and the elasto-plastic constitutive relation. As a result, we included a subsection "3.1 Rate formulation and elasto-plasticity", which provides the reader with a detailed presentation of the constitutive relation we implemented in the solver. In addition, we focused our performance analysis for the selected case of the elasto-plastic granular collapse considering both the number of iteration per second and the number of floating-point operation per seconds with respect to the total number of material points in the system. Furthermore, we also decided to restrict our analysis to the total number of material points for two main reasons we detailed in the preamble of the Result section, as this was suggested by the second referee.

**Comment # 1**  The introduction to the paper appears to have picked a random selection of MPM articles rather than focusing on articles that look at the numerical implementation of the method. The introduction should be made more coherent and focused.

**Reply # 1**  We acknowledge the lack of coherence of the introduction section. Consequently, we focus our introduction toward the numerical implementation of MPM and add references which focus on numerical implementation of MPM and FEM, since both share common grounds. Especially, we present with greater details the concepts of vectorisation, blocking, overheads and RAM-to-cache communication issues.

**Change # 1**  (L31-L58) MATLAB$^\copyright$ allows a rapid code prototyping but, at the expense of significantly lower computational performances than compiled language. An efficient MATLAB implementation of FEM called MILAMIN (Million a Minute) was proposed by Dabrowski et al. (2008) that was capable of solving two-dimensional linear problems with one million unknowns in one minute on a modern computer with a reasonable architecture. The efficiency of the algorithm lies on a combined use of vectorised calculations with a technique called blocking. MATLAB uses the Linear Algebra PACKages (LAPACK), written in Fortran, to perform mathematical operations by calling Basic Linear Algebra Subroutines (BLAS, Moler (2000)). The latter results in an overhead each time a BLAS call is made. Hence, mathematical operations over a large number of small matrices should be avoided and, operations on fewer and larger matrices preferred. This is

a typical bottleneck in FEM when local stiffness matrices are assembled during the integration point loop within the global stiffness matrix. Dabrowski et al. (2008) proposed an algorithm, in which a loop reordering is combined with operations on blocks of elements to address this bottleneck. However, data required for a calculation within a block should entirely resides in the CPUs cache. Otherwise, an additional time is spent on the RAM-to-cache communication and the performance decreases. Therefore, an optimal block size exists and, is solely defined by the CPU architecture. This technique of vectorisation combined with blocking significantly increases the performance.

More recently, Bird et al. (2017) extended the vectorised and blocked algorithm presented by Dabrowski et al. (2008) to the calculation of the global stiffness matrix for Discontinuous Galerkin FEM considering linear elastic problems using only native MATLAB functions. Indeed, the optimisation strategy chosen by Dabrowski et al. (2008) also relied on non-native MATLAB functions, e.g., `sparse2` of the SuiteSparse package (Davis, 2013). In particular, Bird et al. (2017) showed the importance of storing vectors in a column-major form during calculation. Mathematical operations are performed in MATLAB by calling LAPACK, written in FORTRAN, in which arrays are stored in column-major order form. Hence, element-wise multiplication of arrays in column-major form is significantly faster and thus, vectors in column-major form are recommended, whenever possible. Bird et al. (2017) concluded that vectorisation alone results in a performance increase between 13.7 and 23 times, while blocking only improved vectorisation by an additional 1.8 times. OSullivan et al. (2019) recently extended the works of Bird et al. (2017); Dabrowski et al. (2008) to optimised elasto-plastic codes for Continuous Galerkin (CG) or Discontinuous Galerkin (DG) methods. In particular, they proposed an efficient native MATLAB function, i.e., `accumarray()`, to efficiently assemble the internal force vector. Such function constructs an array by accumulation. More generally, OSullivan et al. (2019) reported a performance gain of x25.7 when using an optimised CG code instead of an equivalent non-optimised code.

**Comment # 2**    The authors have not picked up on any of the papers that extend the MILAMIN approach. In particular the authors should refer to the following paper that extends the MILAMIN ideas to non-linear problems: OSullivan, S, Bird, R.E., Coombs, W.M. and Giani, S. (2019). Rapid non-linear finite element analysis of continuous and discontinuous Galerkin methods in MATLAB. Computers and Mathematics with Applications 78(9): 3007-3026. There are others and the authors should review the appropriate literature.

**Reply # 2**    Indeed, we were not aware of further extension to the MILAMIN approach. We reviewed the reference proposed and found common approach concerning vectorization, i.e., the use of the accumarray function to accumulate internal force contributions to nodes. Additionally, we reviewed the appropriate literature and consequently update the introduction section. As such, our reply follows the previous statement of our first reply.

**Change # 2**    See Change # 1.

**Comment # 3**    If performance is the focus, why use MATLAB? Why not adopt one of the existing MPM codes that are written in compiled code which will always be faster than MATLAB? Such as: jixiefx.com, github.com/yuanminghu/taichi_mpm, cimne.com/kratos/, github.com/nairnj/nairn-mpm-fea, github.com/cbgeo/mpm/, sourceforge.net/p/mpmgimp, github.com/xzhang66/MPM3D-F90. There may be others.

**Reply # 3**    It is true that MPM codes written in compiled codes will always be faster than MATLAB. Our point is to show that one can use MATLAB to produce prototyped code with a decent computational performance. However, our true concern is to propose an efficient and vectorised code, which could be more easily translated into the C CUDA language, or, at least, which partially minimizes the hurdles of the syntax translation while preserving the algorithm. As such, we clearly state such concern in the revised manuscript.

**Change # 3**   (L68-70) The vectorisation of MATLAB functions is also crucial for a straight transpose of the solver to a more efficient language, such as the C-CUDA language, which allows the parallel execution of computational kernels of graphics processing units (GPUs).

(L558-560) The vectorisation activities we performed provide a fast and efficient MATLAB-based MPM solver. Such vectorised code could be transposed to a more efficient language, such as the C-CUDA language, that is known to efficiently take advantage of vectorised operations.

**Comment # 4**   The authors mention an implicit implementation but do not present any results in terms of the speed gains of the implicit vectorised algorithm. These should be included and the vectorised algorithm explained.

**Reply # 4**   It is true that i) we do not present any results related to speed gain and, ii) the implicit implementation is not explained. Our concern was to show that the error saturation we reported for the explicit implementation was due to the explicit formulation. Consequently, an implicit implementation under the vectorisation framework of the explicit formulation should converge without any saturation error. We showed it did. However, we can include in the Supplementary Material of the paper a description of the vectorization for the implicit implementation if needed. But up to now, we decided not to included any description of an implicit MPM implementation. We prefer to directly refer to the relevant literature.

**Change # 4**   (L111-118) Additionally, we implemented a CPDI/CPDI2q version (in an explicit and quasi-static implicit formulation) of the solver. However, in this paper, we do not present the theoretical background of the CPDI variant nor the implicit implementation of a MPM-based solver. Therefore, interested readers are referred to the original contributions of Sadeghirad et al. (2013, 2011) and Acosta et al. (2020); Charlton et al. (2017); Iaconeta et al. (2017); Beuth et al. (2008); Guilkey and Weiss (2003), respectively. Regarding the quasi-static implicit implementation, we strongly adapted our vectorisation strategy to some aspects of the numerical implementation proposed by Coombs and Augarde (2020) in the MATLAB code AMPLE v1.0. However, we did not consider blocking, because our main concern for performance is on the explicit implementation.

**Comment # 5**   CPDI2 methods suffer from issues associated with domain distortion and are not suitable for problems involving large shear/rotation. This point should be acknowledged, see for example: Wang, L., Coombs, W.M. , Augarde, C.E. , Cortis, M. Charlton, T.J. , Brown, M.J. Knappett, J., Brennan, A. Davidson, C. Richards, D. & Blake, A. (2019). On the use of domain-based material point methods for problems involving large distortion. Computer Methods in Applied Mechanics and Engineering 355: 1003-1025.

**Reply # 5**   It is true that CPDI2q suffers from domain distortion, as shown in the cited reference by the referee. We clarified this in the revised manuscript by introducing a new subsection "2.2 Domain-based material point method variants" (L119). In general, we discussed in greater details advantages and flaws of domain-based MPM.

**Change # 5**   (L120-151) Domain-based material point method variants could be treated as two distinct groups:

- The material point's domain is a square for which the deformation is always aligned with the mesh axis, i.e., a non-deforming domain uGIMPM (Bardenhagen and Kober, 2004) or, a deforming domain cpGIMPM (Wallstedt and Guilkey, 2008), the latter being usually related to a measure of the deformation, e.g., the determinant of the deformation gradient.

- The material point's domain is either a deforming parallelogram for which its dimensions are specified by two vectors, i.e., CPDI (Sadeghirad et al., 2011), or a deforming quadrilateral solely defined by its corners, i.e., CPDI2q (Sadeghirad et al., 2013). However, the deformation is not necessarily aligned with the mesh anymore.

We first focus on the different domain updating methods for GIMPM. Four domain updating methods exists: i) the domain is not updated, ii) the deformation of the domain is proportional to the determinant of the deformation gradient $\det(F_{ij})$ (Bardenhagen and Kober, 2004), iii) the domain lengths $l_p$ are updated accordingly to the principal component of the deformation gradient $F_{ii}$ (Sadeghirad et al., 2011) or, iv) are updated with the principal component of the stretch part of the deformation gradient $U_{ii}$ (Charlton et al., 2017). Coombs et al. (2020) highlighted the suitability of generalised interpolation domain updating methods accordingly to distinct deformation modes. Four different deformation modes were considered by Coombs et al. (2020): simple stretch, hydrostatic compression/extension, simple shear and, pure rotation. Coombs et al. (2020) concluded the following:

- Not updating the domain is not suitable for simple stretch and hydrostatic compression/extension.

- A domain update based on $\det(F_{ij})$ will results in an artificial contraction/expansion of the domain for simple stretch.

- The domain will vanish with increasing rotation when using $F_{ii}$.

- The domain volume will change under isochoric deformation when using $U_{ii}$.

Consequently, Coombs et al. (2020) proposed a hybrid domain update inspired by CPDI2q approaches: the corners of the material point domain are updated accordingly to the nodal deformation but, the midpoints of the domain limits are used to update domain lengths $l_p$ to maintain a rectangular domain. Even tough Coombs et al. (2020) reported an excellent numerical stability, the drawback is to compute specific basis functions between nodes and material point's corners, which has an additional computational cost. Hence, we did not selected this approach in this contribution.

Regarding the recent CPDI/CPDI2q, Wang et al. (2019) investigated the numerical stability under stretch, shear and torsional deformation modes. CPDI2q was found to be erroneous in some case, especially when torsion mode is involved, due to distortion of the domain. In contrast, CPDI and even sMPM performed better in modelling torsional deformations. Even tough CPDI2q can exactly represent the deformed domain (Sadeghirad et al., 2013), care must be taken when dealing with very large distortion, especially when the material has yielded, which is common in geotechnical engineering (Wang et al., 2019).

Consequently, the domain-based method as well as the domain updating method should be carefully chosen accordingly to the deformation mode expected for a given case. The latter will be always specify in the following and, the domain update method will be clearly stated.

**Comment # 6**   What large deformation formulation has been used in this paper in terms of stresses and strains? Logarithmic strains and Kirchhoff stresses? Explain and justify the large deformation framework.

**Reply # 6**   The large deformation formulation is based on the Jaumann stress rate formulation. It is a widely accepted deformation framework, see Huang et al. (2015); Bandara et al. (2016); Wang et al. (2016c,b) and many others. We think for an explicit implementation with a sufficiently small time step, such deformation formulation stands to reason. As such, we introduced a specific subsection "3.1 Rate formulation and elasto-plasticity".

**Change # 6**   (L156-163) The large deformation framework in a linear elastic continuum requires an appropriate stress-strain formulation. One approach is based on the finite deformation framework, which relies on a linear relationship between elastic logarithmic strains and Kirchoff stresses (Coombs et al., 2020; Gaume et al., 2018; Charlton et al., 2017). In this study, we adopt another approach, namely, a rate dependent formulation using the Jaumann stress rate (e.g. Huang et al. 2015; Bandara et al. 2016; Wang et al. 2016c,b). This formulation provides an objective (invariant by rotation or frame-indifferent) stress rate measure (de Souza Neto et al., 2011) and is simple to implement. The Jaumann rate of the Cauchy stress is defined as

$$\frac{\mathcal{D}\sigma_{ij}}{\mathcal{D}t} = \frac{1}{2}C_{ijkl}\left(\frac{\partial v_l}{\partial x_k} + \frac{\partial v_k}{\partial x_l}\right), \tag{1}$$

where $C_{ijkl}$ is the fourth rank tangent stiffness tensor and $v_k$ is the velocity. Thus, the Jaumann stress derivative can be written as

$$\frac{\mathcal{D}\sigma_{ij}}{\mathcal{D}t} = \frac{\mathrm{D}\sigma_{ij}}{\mathrm{D}t} - \sigma_{ik}\omega_{jk} - \sigma_{jk}\omega_{ik}, \tag{2}$$

where $\omega_{ij} = (\partial_i v_j - \partial_j v_i)/2$ is the vorticity tensor and $\mathrm{D}\sigma_{ij}/\mathrm{D}t$ denotes the material derivative

$$\frac{\mathrm{D}\sigma_{ij}}{\mathrm{D}t} = \frac{\partial \sigma_{ij}}{\partial t} + v_k \frac{\partial \sigma_{ij}}{\partial x_k}. \tag{3}$$

**Comment # 7**  How has the plasticity algorithm been implemented within the large deformation framework? The authors mention a prediction/correction type algorithm so how have they recovered the additive decomposition of the elastic and plastic strains from the multiplicative decomposition of the deformation gradient? Critical details are missing here.

**Reply # 7**  It is true that critical details are missing concerning the plasticity algorithm we selected. We added more details concerning the way plasticity is handle in the revised manuscript. More specifically, we used the prediction/correction algorithm combined with the Jaumann stress rate formulation which allow us to afford the decomposition of the elastic and plastic strain from the decomposition of the deformation gradient, as done in the MATLAB code AMPLE. We present in detail the general implementation of plastic flow accordingly to Simpson (2017) in the subsection "3.1 Rate formulation and elasto-plasticity".

**Change # 7**  (L164-184) Plastic deformation is modelled with a pressure dependent Mohr-Coulomb law with non-associated plastic flow, i.e., both the dilatancy angle $\psi$ and the volumetric plastic strain $\epsilon_v^p$ are null (Vermeer and De Borst, 1984). We have adopted the approach of Simpson (2017) for a two dimensional linear elastic, perfectly plastic (elasto-plasticity) continuum because of its simplicity and its ease of implementation. The yield function is defined as

$$f = \tau + \sigma \sin\phi - c\cos\phi, \tag{4}$$

where $c$ is the cohesion and $\phi$ the angle of internal friction,

$$\sigma = (\sigma_{xx} + \sigma_{yy})/2, \tag{5}$$

and

$$\tau = \sqrt{(\sigma_{xx} - \sigma_{yy})^2/4 + \sigma_{xy}^2}. \tag{6}$$

The elastic state is defined when $f < 0$. However when $f > 0$, plastic state is declared and stresses must be corrected (or scaled) to satisfy the condition $f = 0$, since $f > 0$ is an inadmissible state. Simpson (2017) proposed the following simple algorithm to return stresses to the yield surface,

$$\sigma_{xx}^\star = \sigma + (\sigma_{xx} - \sigma_{yy})\beta/2, \tag{7}$$

$$\sigma_{yy}^\star = \sigma - (\sigma_{xx} - \sigma_{yy})\beta/2, \tag{8}$$

$$\sigma_{xy}^\star = \sigma_{xy}\beta, \tag{9}$$

where $\beta = (\mid c\cos\phi - \sigma\sin\phi \mid)/\tau$, and $\sigma_{xx}^\star$, $\sigma_{yy}^\star$ and $\sigma_{xy}^\star$ are the corrected stresses, i.e., $f = 0$.

A similar approach is used to return stresses when considering a non-associated Drucker-Prager plasticity (see Huang et al. (2015) for a detailed description of the procedure). In addition, their approach allows also to model associated plastic flows, i.e., $\psi > 0$ and $\epsilon_v^p \neq 0$.

**Comment # 8**  How are boundary conditions imposed in this model?

**Reply # 8**   Boundary conditions can be difficult to impose thoroughly in MPM, especially when considering non-aligned boundary conditions with respect to the background mesh, as mentioned in Cortis, M., Coombs, W., Augarde, C., Brown, M., Brennan, A., Robinson, S. Imposition of essential boundary conditions in the material point method. International Journal for Numerical Methods in Engineering. 113: 130-152. This is the main reason why we choose numerical problem and setup in which boundary conditions could be directly applied on the background mesh. We mentioned this important point of aligned boundary conditions in the revised manuscript in the new subsection "3.4 Initial settings and adaptive time step".

**Change # 2**   (L315-319) In this contribution, Dirichlet boundary conditions are resolved directly on the background mesh, as in the standard finite element method. This implies that boundary conditions are resolved only in contiguous regions between the mesh and the material points. Deviating from this contiguity or having the mesh not aligned with the coordinate system requires specific treatments for boundary conditions (Cortis et al., 2018). Furthermore, we ignore the external tractions as their implementation is complex.

**Comment # 9**   Figure 6 - what causes the step in the red line around $10^3$ material points? Is there a shift in terms of the cost within the algorithm?

**Reply # 9**   Thank you for this comment. We started an investigation in terms of flops and L2 cache concerning this step around $10^3$ material points and we just concluded this shift is due to the overhead and RAM-to-cache transfer. As mentioned further by the referee, overheads are a major concern in MATLAB. This step typically occurs when the block of data send from RAM to the CPU L2 cache exceeds its maximum capacity. MATLAB can no longer treat the information as a contiguous block of data sent once and start swapping from RAM to cache and back and forth. This costs an additional computational expense which results in this step in Fig. 6 in the original paper (see 1).

[Figure]

Figure 1: a) Floating-point operation per second in million and, b) linear increase of RAM usage with an increasing number of material point. The limit when the allocated space in RAM reaches the L2 cache size (1024kB) corresponds to the drop of FLOPS in FIG I a).

Thanks to the referees comment, we now come with a different approach to vectorise the matrix multiplication problem which is faster than the previous one but still suffers from overhead due RAM-to-cache communication. For this particular problem, we report a rather significant difference between the two vectorisation variants. This is particularly obvious when we observe the difference in terms of Mflops: flops are at least twice higher than the former implementation for the matrix multiplication (see Fig. 2). In the earlier version of the solver, the matrix multiplication problem was handle by the MATLAB instruction depicted in line 6 in the code listing 1. However, the function `permute( )` is less efficient than using the function `reshape( )`. Hence, we found out it was more efficient to use the MATLAB instruction in line 8 in the code listing 1 instead. However, these two variants give the exact same result as the standard iterative matrix multiplication in line 3 in the code listing 1.

Even tough it does not ensure optimal RAM-to-cache communications, it increased rather significantly the performance compared to the previous version. Consequently, we replaced the previous vectorisation by this new technique and make the necessary changes in our revised paper. It also implied to modify results and observations in the computational efficiency section of the paper. Therefore, the actual version of the solver is v1.1. This has been changed wherever necessary in the revised manuscript. However, we also

[Figure]

Figure 2: a) Floating-point operation per second (in million) for old (dashed lines) and new (continuous lines) vectorisation frameworks and, b) wall-clock time (in ms) for an increasing number of material point $n_p$ for an iterative calculation, the old and the new vectorisation frameworks.

mentioned the drop of efficiency due to overheads. In addition, this computational cost due to RAM-to-cache communication would be an interesting improvement of a future implementation of a vectorised MPM code in MATLAB.

```matlab
%% MATRIX MULTIPLICATION USING FOR LOOP
for k = 1:np(p)
    loop_Bs(:,k) = (B(:,:,k))'*(s(:,k))
        ;% matrix multiplication operator *
end
%% MATRIX-VECTOR MULTIPLICATION USING VECTORISATION
vect_Bs = squeeze(sum(permute(B ,[2 1 3]).*...
        repmat((permute(s',[3 2 1])) ,nDoF,1),2))
    ;% old matrix-vector multiplication
vect_Bs = (squeeze(sum(B.*repmat(reshape(s,3,1,np(p)),1,nDoF)
    ,1)));% new matrix-vector multiplication
```

Listing 1: Original (v1.0) and new (v1.1) vectorised solutions of the matrix multiplication problem.

**Change # 9** (L281-288) To illustrate the numerical efficiency of the vectorised multiplication between a matrix and a vector, we have developed an iterative and vectorised solution of $\boldsymbol{B}(\boldsymbol{x}_p)^T\boldsymbol{\sigma}_p$ with an increasing $n_p$ and considering single (4 bytes) and double (8 bytes) arithmetic precision. The wall-clock time increases with $n_p$ with a sharp transition for the vectorised solution around $n_p \approx 1000$, as showed in Fig 6a. The mathematical operation requires more memory than available in the L2 cache (1024 kB under the CPU architecture used), which inhibits cache reuse (Dabrowski et al., 2008). A peak performance of at least 1000 Mflops, showed in Fig. 6b, is achieved when $n_p = 1327$ or $n_p = 2654$ for simple or double arithmetic precision respectively, i.e., it corresponds exactly to 1024 kB for both precisions. Beyond, the performance dramatically drops to approximately the half of the peak value. This drop is even more severe for a double arithmetic precision.

[Figure]

Figure 6: a) Wall-clock time to solve for a matrix multiplication between a multidimensional array and a vector with an increasing number of the third dimension with a double arithmetic precision and, b) number of floating point operations per second (flops) for single and double arithmetic precisions. The continuous line represents the averages value whereas the shaded area denotes the standard deviation.

**Comment # 10** It is well known that MPMs can provide a reasonable global approximation (in terms of force-displacement response) but the computed stress field can be spurious/highly oscillatory due to issues such as locking and/or cell crossing. The authors should present the predicted stress response for the numerical examples presented in the paper and compare them to analytical solutions where available.

**Reply # 10** This is a good suggestion. We present the stress field for selected numerical problems in the revised manuscript, i.e., the elastic compression and the cantilever beam. Additionally, we also present the analytical solution of the stress $\sigma_{yy}$ for the elastic column and its analytical solution.

**Change # 10**

[Figure]

Figure 9: a) Convergence of the error: a limit is reached at error $\approx 2 \cdot 10^{-6}$ for the explicit solver, whereas the quasi-static solution still converges. This was already demonstrated in Bardenhagen and Kober (2004) as an error saturation due to the explicit scheme, i.e., the equilibrium is never resolved. b) The stress $\sigma_{yy}$ along the $y$-axis predicted at the deformed position $y_p$ by the CPDI2q variant is in good agreements with the analytical solution for a refined mesh.

[Figure]

Figure 13: Finite deformation of the material point domain and vertical Cauchy stress $\sigma_{yy}$ for CPDI, i.e., a) & b), and for cpGIMPM, i.e., c) & d). The CPDI variant gives a better and contiguous description of the material point's domain and a slightly smoother stress field, compared to the cpGIMPM variant, which is based on the stretch part of the deformation gradient.

**Comment # 11**  The elastic column problem will only have around 2 % deformation in terms of the deformed to original height so this problem is not a good test of the convergence of MPMs. The authors refer to the work of Coombs et al. for this problem but they show convergence for the case where the column compresses to around 50 % of its initial height. Run the convergence for the case with a Youngs modulus of 10kPa, not 1MPa.

**Reply # 11**  The referee is right; the elastic column will only have a small vertical deformation. We run the convergence analysis with lower elastic modulus as proposed to achieve a greater deformation of the column.

**Change # 11**  (L360-361) The material has a Young's modulus $E = 1 \cdot 10^4$ Pa and a Poisson's ratio $\nu = 0$ with a density $\rho = 80$ kg m$^{-3}$.

**Comment # 12**   How have the authors avoided volumetric locking when using isochoric plastic flow? What do the pressure distributions look like through the deformed domains?

**Reply # 12**   We did not avoid volumetric locking when using isochoric plastic flow. As a result, the pressure field experience severe oscillations for isochoric plastic deformations. It is true that any MPM variant suffers from volumetric locking. As such, we now clearly state in the revised manuscript that we did not treat the oscillation of the pressure field due to volumetric locking. We also discuss in detail the issue of volumetric locking in the Discussion section and share potential ways to regularize the pressure field, especially when ischoric plastic deformations are involved.

**Change # 12**   (L513-517) This particular case of isochoric plastic deformations rises the issue of volumetric locking. In the actual implementation, no regularization techniques are considered. As a result, the pressure field experience severe locking for isochoric plastic deformations. One way to overcome locking phenomenons would be to implement the regularization technique initially proposed by Coombs et al. (2018) for quasi-static sMPM and GIMPM implementations.

**Comment # 13**   The various comments on the use of cpGIMP and uGIMP are confused and misleading. Just because an analysis is stable it does not mean that it is "appropriate" as the results may be meaningless. The authors cite the work of Coombs to justify the use of the stretch to update the domains, however this technique does not work for problems involving simple shear type deformation, refer to Table 2 of: Coombs, WM, Augarde, CE, Brennan, AJ, Brown, MJ, Charlton, TJ, Knappett, JA, Ghaffari Motlagh, Y & Wang, L (2020). On Lagrangian mechanics and the implicit material point method for large deformation elastoplasticity. Computer Methods in Applied Mechanics and Engineering 358: 112622. The authors later adopt a determinant of the deformation gradient-type updating method but this has been shown to not converge for simple compression problems as the domains artificially shrink in the non-compressed direction and overlap in the compressed direction (see Figure 8 from the above reference). For example, the statement on lines 76-77 is not correct, or rather it is only correct for some types of problem.

**Reply # 13**   We agree that this various comment all around the manuscript were confused and misleading. As such, we elaborated the presentation of domain-based MPMs and correctly refer to advantages and flaws for each of these domain-based variants. Consequently, we included a new subsection in the MPM overview section, e.g., "2.2 Domain-based material point method variants".

   Regarding the determinant of the deformation gradient-type update, the referee comment is absolutely true. However, we selected such approach since it gave the most stable numerical result. But, we clearly mentioned the problem of artificial shrinking and domain overlaps associated with this variant in the revised paper.

**Change # 14**   See Change # 5.

**Comment # 14**   The speed gains of MILAMIN come from the combination of blocking and vectorisation. Have both techniques been used in this paper? If so, what are the relative speed gains from the different sources? How does the speed gain change with different block sizes?

**Reply # 14**   We did use vectorisation but we did not use blocking. It is true that in MILAMIN the speed gain come from a combination between vectorisation and blocking. However, such technique is applied in assembling local stiffness matrix in the global stiffness matrix, which does not appear in an explicit MPM formulation. Hence, the technique of blocking was disregarded for that concern. We acknowledge this statement in the introduction section.

**Change # 14**   (L64-67) We did not consider the blocking technique initially proposed by Dabrowski et al. (2008) since an explicit formulation in MPM excludes the global stiffness matrix assembly procedure. The performance gain mainly comes from the vectorisation of the algorithm, whereas blocking has a less significant impact over the performance gain, as stated by Bird et al. (2017).

**Comment # 15**   The authors have not explained by MATLAB is inefficient when working with small amounts of data. This point should be discussed with reference to CPU cache size and RAM-to-cache overheads.

**Reply # 15**   We discussed this point explicitly when presenting the new vectorisation framework of the matrix multiplication problem we mentioned previously in our response. In addition, we also discuss this issue in the section "5. Discussion".

**Change # 15**   (L37-39) Hence, mathematical operations over a large number of small matrices should be avoided and, operations on fewer and larger matrices preferred. This is a typical bottleneck in FEM when local stiffness matrices are assembled during the integration point loop within the global stiffness matrix.

(L284-285) The mathematical operation requires more memory than available in the L2 cache (1024 kB under the CPU architecture used), which inhibits cache reuse (Dabrowski et al., 2008).

(538-542) An iterative implementation would require multiple nested for-loops and a larger number of operations on smaller matrices, which increase the number of BLAS calls, thus inducing significant BLAS overheads and decreasing the overall performance of the solver. This is limited by a vectorised code structure. However and as showed by the matrix multiplication problem, the L2 cache reuse is the limiting factor and, it ultimately affects the peak performance of the solver due to these numerous RAM-to-cache communications for larger matrices.

**Comment # 16**   It would be more appropriate to present the speed gains in terms of flops. There should be a peak in performance if you consider large enough problems.

**Reply # 16**   Thank you for this relevant suggestion. As mentioned previously in our reply, we already started to investigate the speed gain of the matrix-vector operation problem. We extended this point to our computational performance analysis, which greatly benefited from this. Our performance analysis focused on both the number of iterations per second and the number of floating-point operations per second with respect to the total number of material points. As stated by the referee, a peak performance is reached and then, a residual performance is further resolved as the total number of material points increases.

**Change # 16**   (L473-482) The performance of the solver is demonstrated in Fig. 18. A peak performance of $\approx 900$ Mflops is reached, as soon as $n_p$ exceeds 1000 material points and, a residual performance of $\approx 600$ Mflops is further resolved (for $n_p$ approximately 50000 material points). Every functions provide an even and fair contribution on the overall performance, except the function `constitutive.m` for which the performance appears delayed or shifted. First of all, this function treats the elasto-plastic constitutive relation, in which the dimensions of the matrices are smaller when compared to the other functions. Hence, the amount of floating point operations per second is lower compared to other functions, e.g., `p2Nsolve.m`. This results in less performance for an equivalent number of material points. It also requires a greater number of material points to increase the dimensions of the matrices in order to exceed the L2 cache maximum capacity.

This considerations provide a better understanding of the performance gain of the vectorised solver showed in Fig. 19: the gain increases and then, reaches a plateau and ultimately, decreases to a residual gain. This is directly related to the peak and the residual performances of the solver showed in Fig. 18.

[Figure]

Figure 18: Number of floating point operation per seconds (flops) with respect to the total number of material point $n_p$ for the vectorised implementation. The discontinuous lines refer to the functions of the solver, whereas the continuous line refer to the solver. A peak performance of 900 Mflops is reached by the solver for $n_p > 1000$ and, a residual performance of 600 Mflops is further resolved for an increasing $n_p$.

[Figure]

Figure 19: Number of iterations per second with respect to the total number of material point $n_p$. The greatest performance gain is reached around $n_p = 1000$, which is related to the peak performance of the solver (see Fig. 18). The gains corresponding to the peak and residual performances are 46 and 28 respectively.

**Comment # 17**  Figure 2 is misleading as the GIMP domain is fully inside a single element and will have the same connectivity as the standard MP case.

**Reply # 17**  We agree that Figure 2 was misleading. Consequently, we replaced the previous Figure 2 by a new one in the revised manuscript. In addition, we also included the nodal connectivity for the CPDI2q variant.

**Change # 17**

[Figure]

Figure 2: Nodal connectivities of a) standard MPM, b) GIMPM and c) CPDI2q variants. The material point's location is marked by the blue cross. Note that for sMPM (and similarly BSMPM) the particle domain does not exist, unlike GIMPM or CPDI2q (the blue square enclosing the material point). Nodes associated with the material point are denoted by filled blue squares, and the element number appears in green in the centre of the element. For sMPM and GIMPM, the connectivity array between the material point and the element is p2e and, the array between the material point and its associated nodes is p2N. For CPDI2q, the connectivity array between the corners (filled red circles) of the quadrilateral domain of the material point and the element is c2e and, the array between the corners and their associated nodes is c2N.

**Comment # 18**  Line 162 - the authors mention a 30 % speed up - what problem/size of problem/number of points, etc?

**Reply # 18**  We mentioned a 30 % speed up at Line 162 in the original manuscript for the calculation of the shape function. We carried out a more thoroughly analysis of such speed up and observe a steady gain regardless of the number of material points or the number of element. Consequently, we simply mention this within the text. However, we also noted the sentence was not clear enough. Therefore, we made our point more straight forward in indicating that such speed up was only valid for the calculation of the shape function during a computational cycle and not over the whole solver.

**Change # 18**  (L247-249) The performance gain is significant between the two approaches, i.e., an intrinsic 30 % gain over the wall-clock time of the basis functions and derivatives calculation. We observe an invariance of such gain with respect to the initial number of material point per element or to the mesh resolution.

**Comment # 19**  Where are material points located within the elements when the problems are set up?

**Reply # 19**  Regarding the initial location of material points within elements, we preferred a regular distribution of material point within elements. Considering 4 material points per element, their location (in local coordinate) would be [-0.5; 0.5] along x and y direction. Hence, we decided to simply mention that we only consider a regular mesh spacing with a uniform distribution of material points per initially populated element. As suggested by the second referee, this was included in the new subsection "3.3 Initial settings and adaptive time step".

**Change # 19** (L312-314) Regarding the initial setting of the background mesh of the demonstration cases further presented, we select a uniform mesh and a regular distribution of material points within the initially populated elements of the mesh. Each element is evenly filled with 4 material points, e.g., $n_{pe} = 2^2$, unless otherwise stated.

**Comment # 20** Some key information is missing from the numerical analysis, such as time step sizes and total times which would make it impossible to reproduce the results. This information must be added.

**Reply # 20** During the revision process of our contribution, we decided to consider an adaptive time step. Hence, we clearly mention this in the revised manuscript and we provide the reader with the CFL number $C$ we used for every demonstration case. Whenever relevant, we also mention the minimal and maximal time steps obtained given elastic properties of the material and the CFL number chosen and we also specify the total simulation time.

**Change # 20** (L320-328) As explicit time integration is only conditionally stable, any explicit formulation requires a small time step $\Delta t$ to ensure numerical stability (Ni and Zhang, 2020), e.g., smaller than a critical value defined by the Courant-Friedrich-Lewy (CFL) condition. Hence, we employ an adaptive time step (de Vaucorbeil et al., 2020), which considers the velocity of the material points. The first step is to compute the maximum wave speed of the material using (Zhang et al., 2016; Anderson Jr, 1987)

$$(c_x, c_y) = \{\max_p(V + |(v_x)_p|), \max_p(V + |(v_y)_p|)\}, \tag{10}$$

where the wave speed is $V = ((K + 4G/3)/\rho)^{\frac{1}{2}}$, $K$ and $G$ are the bulk and shear moduli respectively, $\rho$ is the material density, $(v_x)_p$ and $(v_y)_p$ are the material point velocity components. $\Delta t$ is then restricted by the CFL condition as followed:

$$\Delta t = \alpha \min\left(\frac{h_x}{c_x}, \frac{h_y}{c_y}\right), \tag{11}$$

where $\alpha \in [0; 1]$ is the time step multiplier, and $h_x$ and $h_y$ are the mesh spacings.

**Comment # 21** Incomplete sentence on line 359.

**Reply # 21** That is correct, sentence on line 359 was completed in the revised manuscript.

**Change # 21** (L497-498) In this contribution, a fast and efficient explicit MPM solver is proposed that considers two variants (e.g., the uGIMPM/cpGIMPM and the CPDI/CPDI2q variants).

**2 Referee #2**

This manuscript presents a vectorized material point method for MATLAB and quantifies the increase in computational efficiency gained from using vectorized rather than iterative code. To my knowledge, this paper provides the only formal analysis of the performance gains from vectorizing MPM code. The presented vectorization approach could be easily implemented within existing and future MPM models. However, as already thoroughly noted by Referee #1, many details of the algorithms, setup of simulations, and numerical analysis are missing. I would like to add the following comments to those already given.

**Comment # 1** I found the structure of the paper to be confusing, especially in Section 4, where results from five test cases are reported. These test cases are all nearly exact reproductions of previously-published work. The first two test cases the elastic compaction of a column (Section 4.1) and elastic cantilever beam (Section 4.2) appear to solely serve as benchmark examples for verification of the MPM model, though this is not clearly stated. The third test case elasto-plastic column collapse (Section 4.3) also appears to serve as further verification of the model until it is used again in Section 4.4, where the main results of the paper concerning the computational efficiency gained from vectorization are presented. A fourth test case (collision

of two elastic disks) is also presented in 4.4 for further analysis of computational efficiency. The final test case (elasto-plastic landslide) is then presented in Section 4.5, which seems to serve the dual purpose of further model verification and a geomechanical application.

**Reply # 1** It is true that the structure might be confusing. We focus on the section "4. Results" and made our point clearer by adding, as suggested in the following by the referee, a quick introduction to this section. We think it improves the readability of this section.

Overall, the reorganised section "4. Results" now reads as:

1. Results

   (a) Validation of the solver and numerical efficiency

      i. Convergence: elastic compaction under self-weight of a column
      ii. Large deformation: the elastic cantilever beam problem
      iii. Application: the elasto-plastic slumpin dynamics

   (b) Computational performance

      i. Iterative and vectorised elasto-plastic collapses
      ii. Comparison between Julia and MATLAB

We also clearly state our motivation of the exact reproduction of previously-published works. We think it is also important to showcase the efficiency of our solver with benchmarks from past studies. However, we reorganized the subsection in which computational performances are showed, in order to make our point more obvious to the reader.

**Change # 1** (L330-344) In this section, we first demonstrate our MATLAB-based MPM solver to be efficient in reproducing results from other studies, i.e., the compaction of an elastic column (Coombs et al., 2020) (e.g., quasi-static analysis), the cantilever beam problem (Sadeghirad et al., 2011) (e.g., large elastic deformation) and an application to landslide dynamics (Huang et al., 2015) (e.g., elasto-plastic behaviour). Then, we present both the efficiency and the numerical performances for a selected case, e.g., the elasto-plastic collapse. We conclude and compare the performances of the solver with respect to the specific case of an impact of two elastic disks previously implemented in a Julia language environment by (Sinaie et al., 2017).

Regarding the performance analysis, we investigate the performance gain of the vectorised solver considering a double arithmetic precision with respect to the total number of material point because of the following reasons: i) the mesh resolution, i.e., the total number of elements $n_{el}$, influences the wall-clock time of the solver by reducing the time step due to the CFL condition hence increasing the total number of iterations. In addition, ii) the total number of material points $n_p$ increases the number of operations per cycle due to an increase of the size of matrices, i.e., the size of the strain-displacement matrix depends on $n_p$ and not on $n_{el}$. Hence, $n_p$ consistently influences the performance of the solver whereas $n_{el}$ determines the wall-clock time of the solver. The performance of the solver is addressed through both the number of floating point operations per second (flops), and by the average number of iteration per second (it/s). The number of floating point operations per second was manually estimated for each function of the solver.

**Comment # 2** The motivation behind most of the test cases is not clear, especially on the first read. Section 4 would benefit from a short introduction (before 4.1) that outlines what test cases were selected and for what purpose.

**Reply # 2** Thank you for this comment. We agree that the motivation behind most of the test cases is not clear. Consequently, we have now written two paragraphs (as a preamble) at the very begining of the section "4. Results" in which we clearly state our motivations of i) reproducing efficiently results from other studies and ii) analyse the computational performance of our solver for two selected cases, i.e., the elasto-plastic granular collapse and the elastic disk impact problem.

**Change # 2**   See Change # 1

**Comment # 3**   It may help readability if the test cases for elasto-plastic column collapse and collision of two elastic disks are separated into an entirely different section from the rest of the examples, as these two test cases provide the main results in the paper regarding the computational efficiency gained by vectorization.

**Reply # 3**   It is a good suggestion and it should even significantly help the readability of the manuscript. We created a new subsection "4.2 Computational performance" composed of two subsubsections, namely "4.2.1 Iterative and vectorised elasto-plastic collapses" and "4.2.2 Comparison between Julia and MATLAB".

**Change # 3**   See Reply # 1 and Change # 1

**Comment # 4**   Many of the statements regarding the effectiveness of cpGIMPM vs. uGIMPM vs. CPDI are misleading or lacking in detail:

**Reply # 4**   We agree with the Referee that the statements related to the effectiveness of uGIMPM, cpGIMPM and CPDI are misleading or lacking in detail. Therefore, we created a new subsection "2.2 Domain-based material point method variants", which summarises the major difference between GIMPM and CPDI (the way the material points domain is updated). We also present the different domain updating methods for GIMPM and we more clearly highlight the suitability of each of these domains updating methods based on the investigation of Coombs, WM, Augarde, CE, Brennan, AJ, Brown, MJ, Charlton, TJ, Knappett, JA, Ghaffari Motlagh, Y & Wang, L (2020). On Lagrangian mechanics and the implicit material point method for large deformation elasto-plasticity. Computer Methods in Applied Mechanics and Engineering 358: 112622, and, Wang, L., Coombs, W.M. , Augarde, C.E. , Cortis, M. Charlton, T.J. , Brown, M.J. Knappett, J., Brennan, A. Davidson, C. Richards, D. & Blake, A. (2019). On the use of domain-based material point methods for problems involving large distortion. Computer Methods in Applied Mechanics and Engineering 355: 1003-1025. Finally, we explicitly state at the end of new 2.2 subsection that the domain-based method as well as the domain update method should be carefully chosen accordingly to the deformation mode expected for a given case. Moreover, the domain update method is clearly stated in each case presented in the section "5. Results". We believe that now this additional 2.2 subsection provides a better overview of both domain-based method and domain updating method.

**Change # 4**   See Change # 5 for the first referee

**Comment # 5**   Line 288: in what way did domain updates based on the deformation gradient result in failure?

**Reply # 5**   The domain updates based on the deformation gradient showed some flaws of GIMP-based implementation, i.e. the domain artificially shrinks under large rotation, which was already demonstrated in Fig.4 in Coombs, WM, Augarde, CE, Brennan, AJ, Brown, MJ, Charlton, TJ, Knappett, JA, Ghaffari Motlagh, Y & Wang, L (2020). On Lagrangian mechanics and the implicit material point method for large deformation elasto-plasticity. Computer Methods in Applied Mechanics and Engineering 358: 112622. We now clearly mention the reason why the simulation in which the domain updates based on the deformation gradient failed for the cantilever beam problem. In addition, we restructured the elasto-plastic collapse. Hence, we now only mentioned that we performed preliminary analyses to select the most suitable domain updating method for the cpGIMPM variant.

**Change # 5**   (L398-399) As indicated in Sadeghirad et al. (2011), the cpGIMPM simulation failed when using the diagonal components of the deformation gradient to update the material point domain, i.e., the domain vanishes under large rotations as stated in Coombs et al. (2020).

(L460-464) We conducted preliminary investigations using either uGIMPM or cpGIMPM variants, the latter with a domain update based either on the determinant of the deformation gradient or on the diagonal

components of the stretch part of the deformation gradient. We concluded the uGIMPM was the most reliable, even tough its suitability is restricted to both simple shear and pure rotation deformation modes (Coombs et al., 2020).

**Comment # 6**    Line 301: [The elasto-plastic MPM solver] demonstrates the inability of the MPM variants based on a domain update (GIMPM or CPDI) to resolve extremely large plastic deformations when relying on the normal components of the deformation gradient or its stretch part to update the material point domain. This is too general of a statement. There are many cases in which these domain updates would work well; for example, if simple shear is minimal and the "stretch" update is used (Coombs et al 2020). A similarly-flawed statement is made in the conclusion section (lines 394-396). Better conclusions regarding GIMPM/CPDI might incorporate the performance gains reported using the vectorization scheme for calculation of the shape functions and the difference in computational efficiency measured for GIMPM vs CPDI.

**Reply # 6**    We agree that the statement on Line 301 is too general. As such, we specified our concern regarding the deformation mode involved avoiding such flawed statements in the revised version of the manuscript, referring explicitly to the work of Coombs, WM, Augarde, CE, Brennan, AJ, Brown, MJ, Charlton, TJ, Knappett, JA, Ghaffari Motlagh, Y & Wang, L (2020). On Lagrangian mechanics and the implicit material point method for large deformation elasto-plasticity. Computer Methods in Applied Mechanics and Engineering 358: 112622. As suggested by the Referee, we now focus our discussion on the performance gain of GIMPM and CPDI variants, which is indeed the main point of our contribution.

**Change # 6**    (L531-550) The computational performance comes from the combined use of the connectivity array `p2N` with the built-in function `accumarray( )` to i) accumulate material point contributions to their associated nodes or, ii) to interpolate the updated nodal solutions to the associated material points. When a residual performance is resolved, an overall performance gain (e.g., the amount of it/s) of 28 is reported. As an example, the functions `p2nsolve.m` and `mapN2p.m` are 24 and 22 times faster than an iterative algorithm when the residual performance is achieved. The overall performance gain is in agreement to other vectorised FEM codes, i.e., OSullivan et al. (2019) reported an overall gain of 25.7 for a optimised continuous Galerkin finite element code.

An iterative implementation would require multiple nested for-loops and a larger number of operations on smaller matrices, which increase the number of BLAS calls, thus inducing significant BLAS overheads and decreasing the overall performance of the solver. This is limited by a vectorised code structure. However and as showed by the matrix multiplication problem, the L2 cache reuse is the limiting factor and, it ultimately affects the peak performance of the solver due to these numerous RAM-to-cache communications for larger matrices. Such problem is serious and, its influence is demonstrated by the delayed response in terms of performance for the function `constitutive.m`. However, we also have to mention that the overall residual performance was resolved only for a limited total number of material points. The performance drop of the function `constitutive.m` has never been achieved. Consequently, we suspect an additional decrease of overall performances of the solver for larger problems.

The overall performance achieved by the solver is higher than expected and, is even higher with respect to what was reported by Sinaie et al. (2017). We demonstrate that MATLAB is even more efficient than Julia, i.e., a minimum 2.86 performance gain achieved compared to a similar Julia CPDI2q implementation. This confirms the efficiency of MATLAB for solid mechanics problems, provided a reasonable amount of time is spent on the vectorisation of the algorithm.

**Comment # 7**    Line 305: As already pointed out by Referee #1, it should be noted that the determinant of the deformation gradient-type GIMPM domain update is problematic for simple compression problems. Have the authors tried updating the GIMPM domains with the corner scheme from Eqs 35-37 in Coombs et al (2020)? Perhaps it would be more robust.

**Reply # 7**    This is a good comment. First, we added a presentation of the suitability of domain updating methods in GIMPM for the possible deformation modes, directly referring to the work of Coombs, WM, Augarde, CE, Brennan, AJ, Brown, MJ, Charlton, TJ, Knappett, JA, Ghaffari Motlagh, Y & Wang, L

(2020). On Lagrangian mechanics and the implicit material point method for large deformation elasto-plasticity. Computer Methods in Applied Mechanics and Engineering 358: 112622. This is important and is lacking up to now. Concerning the investigation of the corner update, we did not have tried to implement it. We decided to quickly investigate this scheme but we figured out it was necessary to compute additional shape functions between material points corners and nodes, similarly to the CPDI variant. Our concern comes from the need to compute these additional quantities, alongside to the shape function relating material points to their associated nodes. This extra cost is significant, especially when our major concern is computational performance. We therefore decided not to pursue such investigation for that reason. We discuss this in the section "5. Discussion"

**Change # 7** (L526-530) We did not selected the domain updating method based on the corners of the domain as suggested in Coombs et al. (2020). This is because such domain updating method necessitates to calculate additional shape functions between the corners of the domain of the material point with their associated nodes. This results in an additional computational cost. Nevertheless, such variant is of interest and should be addressed as well when the computational performances are not the main concern.

**Comment # 8** There does not appear to be any reference to Fig. 1 in the main text. The caption for Fig. 1 also appears to lack any description of panels B and C.

**Reply # 8** Indeed, there was no reference to Fig. 1 in the main text. We added a description of panels B and C in the figure caption as highlighted by the referee and, we refer now to this Fig. 1 in the main text, as mentioned in the following reply # 9.

**Change # 8** (L81-93) Hence, a typical calculation cycle (see Fig. 1) consists of the three following steps (Wang et al., 2016a):

1. A Mapping phase, during which properties of the material point (mass, momentum or stress) are mapped to the nodes.

2. An updated-Lagrangian FEM (UL-FEM) phase, during which the momentum equations are solved on the nodes of the background mesh and, the solution is explicitly advanced forward in time.

3. A Convection phase, during which i) the nodal solutions are interpolated back to the material points, and ii) the properties of the material point are updated.

[Figure]

a) Points to Nodes Projection     b) Nodal solution     c) Nodes to Points Interpolation

Figure 1: Typical calculation cycle of a MPM solver for a homogeneous velocity field, inspired by Dunatunga and Kamrin (2017). a) The continuum (orange) is discretized into a set of Lagrangian material points (red dots), at which state variables or properties (e.g., mass, stress, and velocity) are defined. The latter are mapped to an Eulerian finite element mesh made of nodes (blue square). b) Momentum equations are solved at the nodes and, the solution is explicitly advanced forward in time. c) The nodal solutions are interpolated back to the material points and, their properties are updated.

**Comment # 9** The list of the three steps of a typical MPM cycle at the beginning of Section 3.1 seems misplaced and is somewhat repetitive of the description of MPM in the previous sections. I suspect this list should link with Fig. 1 and may be more appropriately located within Section 2.

**Reply # 9** We agree with the Referee; this list is misplaced. Consequently, we moved it in the section "2. Overview of the Material Point Method (MPM)" within the subsection "2.1 A Material Point Method implementation" and, we directly refer to this list with Fig. 1.

**Change # 9** See Change # 8

**Comment # 10** Fig 2. In the caption, GIMP should be changed to GIMPM to match the rest of the paper. The authors have already fixed the error in Fig. 2b regarding which nodes are associated with the GIMPM domain.

**Reply # 10** Right, we have made the change in the revised paper.

**Change # 10** -

**Comment # 11** Line 39: which numerical considerations from MILAMIN are used? This is not specifically addressed.

**Reply # 11** Due to the comment of the first referee concerning the lack of focus in the introduction section, we partially have rewritten it. We now specify in the section "1. Introduction" the considerations we selected (from MILAMIN but also from the works of Birds et al, 2017 and OSullivan et al, 2019), which are the use of the function `accumarray()` and vectorisation activities mainly to reduce the number of BLAS calls.

**Change # 11** (L64-70) These techniques include the use of `accumarray()`, optimal RAM-to-cache communication, minimum BLAS calls and the use of native MATLAB functions. We did not consider the blocking technique initially proposed by Dabrowski et al. (2008) since an explicit formulation in MPM excludes the global stiffness matrix assembly procedure. The performance gain mainly comes from the vectorisation of the algorithm, whereas blocking has a less significant impact over the performance gain, as stated by Bird et al. (2017).

**Comment # 12** several citations are missing parentheses (e.g. lines 52 and 84).

**Reply # 12** We have fixed these numerous citation problems in the revised paper.

**Change # 12** -

**References**

[revised manuscript text omitted]

constructed for a two-dimensional model, as follows:

$$S_{\underline{In}}(\boldsymbol{x}_p) = S_{\underline{In}}(x_p)S_{\underline{In}}(y_p), \tag{12}$$

for which the derivative is defined as:

$$\nabla S_{\underline{In}}(\boldsymbol{x}_p) = (\partial_x S_{\underline{In}}(x_p)S_{\underline{In}}(y_p), S_{\underline{In}}(x_p)\partial_y S_{\underline{In}}(y_p)). \tag{13}$$

~~Within the GIMPM variant, uGIMPM (undeformed GIMPM) and cpGIMPM (contiguous particle GIMPM) can be chosen to update the material point domain length $l_p^0$, i.e., $l_p = l_p^0$ for the undeformed GIMPM, $l_{i,p} = \det(F_{jk})l_{i,p}^0$ or $l_{i,p} = F_{ii}l_{i,p}^0$ where $F_{ii}$ is the diagonal component of the deformation gradient for the cpGIMPM. However, Charlton et al. (2017) recommend using the diagonal components $U_{ii} = (F_{kj}F_{ki})^{0.5}$ of the stretch part of the deformation gradient instead of the deformation gradient. Hence, the variant we use relies either on the determinant or on the stretch part of the deformation gradient.~~

[revised manuscript text omitted]

**3.3.2   Integration of internal forces**

Another computationally expensive operation for MATLAB© is the mapping (or accumulation) of the material point contributions to their associated nodes. It is performed by the function `p2Nsolve.m` in the workflow of the solver.

The standard calculations for the material point contributions to the lumped mass $m_n$, the momentum $\mathbf{p}_n$, the external $\mathbf{f}_n^e$ and internal $\mathbf{f}_n^i$ forces are given by:

$$m_n = \sum_{p\in n} S_n(\mathbf{x}_p) m_p, \tag{15}$$

$$\mathbf{p}_n = \sum_{p\in n} S_n(\mathbf{x}_p) m_p \mathbf{v}_p, \tag{16}$$

$$\mathbf{f}_n^e = \sum_{p\in n} S_n(\mathbf{x}_p) m_p \mathbf{b}_p, \tag{17}$$

$$\mathbf{f}_n^i = \sum_{p\in n} v_p \mathbf{B}^T(\mathbf{x}_p) \boldsymbol{\sigma}_p, \tag{18}$$

with $m_p$ the material point mass, $\mathbf{v}_p$ the material point velocity, $\mathbf{b}_p$ the body force applied to the material point and $\boldsymbol{\sigma}_p$ the material point Cauchy stress tensor in the Voigt notation.

Once the mapping phase is achieved, the equations of motions are explicitly solved forward in time on the mesh. Nodal accelerations $\mathbf{a}_n$ and velocities $\mathbf{v}_n$ are given by:

$$\mathbf{a}_n^{t+\Delta t} = m_n^{-1}(\mathbf{f}_n^e - \mathbf{f}_n^i), \tag{19}$$

[revised manuscript text omitted]

Our elasto-plastic MPM solver closely follows the implementation of Huang et al. (2015), except our solver relies on an MUSL procedure, whereas Huang et al. (2015) selected the USF procedure. The We also report a quite significant difference in execution time between the CPDI variant compared to the CPDI2q and cpGIMPM variants, i.e., CPDI executes in an average 280.54 it/s whereas both CPDI2q and CPGIMPM execute in an average 301.42 it/s and an average 299.33 it/s, respectively.

**4.1.1 Application: the elasto-plastic slumping dynamics**

We present an application of the MPM solver (vectorised and iterative version) to the case of landslide mechanics. We selected the domain-based CDPI variant since it performs better than the CPDI2q variant in modelling torsional and stretch deformation modes (Wang et al., 2019) coupled to an elasto-plastic problem is solved by i) an elastic trial, ii) corrected by a return mapping

 constitutive model based on a ~~non-associated (the dilation angle $\psi = 0$) perfectly plastic behaviour of the material. Since the MPM variant in Huang et al. (2015) was not clearly stated, we use the uGIMP variant . The reason is the collapse results in extreme deformations, for which a domain update based on the deformation gradient systematically resulted in a failure during the simulation. We conducted preliminary investigations~~

[revised manuscript text omitted]

~~In addition, we also monitor the average time spent on the different functions called during a single execution cycle $\bar{t}_c$ (Fig. ??). When comparing time spent on the functions `p2Nsolve.m` and `mapN2p.m`, we report a speed up of 24 and 22, respectively. This difference is expected since these functions originally necessitate an extensive use of two nested for-loops to calculate i) the material point contribution or ii) the interpolation of updated nodal solutions~~This considerations provide a better understanding of the performance gain of the vectorised solver showed in Fig. 19: the gain increases and then, reaches a plateau and ultimately, decreases to a residual gain. This is directly related to the peak and the residual performances of the solver showed in Fig. 18.

[Figure]

**Figure 19.**  Number of iterations per second with respect to the  total number of material point $n_p$. The greatest performance gain is reached around $n_p = 1000$, which is related to the  peak performance of the  solver  (see Fig. 18). The gains corresponding to the peak and residual performances are 46 and 28 respectively.

**4.2.2 Comparison between Julia and MATLAB**

We compare the computational efficiency of the vectorised CPDI2q  MATLAB implementation and the computational efficiency reported by Sinaie et al. (2017) of a Julia-based implementation of the collision of two elastic disks problem. However, we note a difference between the actual implementation and the one used by Sinaie et al. (2017); the latter is based on a USL variant with a  cut-off algorithm, whereas  the present implementation relies on the MUSL (or double mapping) procedure, which necessitates a double mapping procedure. The initial geometry and parameters are the same as those used in Sinaie et al. (2017). However, the time step is adaptive and, we select a time step multiplier $\alpha = 0.5$. Given the variety of mesh resolution, we do not present minimal and maximal time step values.

Our CPDI2q implementation, in MATLAB R2018a, is,  at least, 2.8 times faster than the Julia implementation proposed by Sinaie et al. (2017) for similar hardware (see Table 1). Sinaie et al. (2017) completed the analysis with an Intel Core i7-6700 (4 cores with a base frequency of 3.40 GHz up to a turbo frequency of 4.00 GHz) with 16 GB RAM, whereas we used an Intel Core i7-4790 with similar specifications (see  Section 2). However, the performance ratio between MATLAB and Julia seems to decrease as the mesh resolution increases.

**4.3**

**Table 1.** Efficiency comparison of the Julia implementation of Sinaie et al. (2017), and the MATLAB-based implementation for the two elastic disk impact problems.

| mesh | $n_{pe}$ | $n_p$ | Julia | MATLAB | Gain |
|---|---|---|---|---|---|
| | | | | Its/s | |
| $20 \times 20$ | $2^2$ | 416 | 132.80 |  450.27 | 3.40 |
| $20 \times 20$ | $4^2$ | 1'624 | 33.37 |  118.45 | 3.54 |
| $40 \times 40$ | $2^2$ | 1'624 | 26.45 |  115.59 | 4.37 |
| $80 \times 80$ | $4^2$ | 25'784 | 1.82 |  5.21 | 2.86 |

610

615

620

625

630

~~The maximal horizontal extent of the slump is smaller for the MPM solver with an M-C model, but the failure surface is in good agreement with the solution reported by Huang et al. (2015). They did not mention any use of a local damping in their implementation, and this can certainly explain the difference in the maximal extent. The local damping forces implemented in our solver should result in a smaller horizontal extent (because of the extra dissipation term introduced) but these should~~

635

**5 Discussion**

[revised manuscript text omitted]

and decreasing the overall performance of the solver. This is limited by a vectorised code structure. However and as showed by the matrix multiplication problem, the L2 cache reuse is the limiting factor and, it ultimately affects the peak performance of the solver due to these numerous RAM-to-cache communications for larger matrices. Such problem is serious and, its influence is demonstrated by the delayed response in terms of performance for the function `constitutive.m`. However, we also have to mention that the overall residual performance was resolved only for a limited total number of material points. The performance drop of the function `constitutive.m` has never been achieved. Consequently, we suspect an additional decrease of overall performances of the solver for larger problems.

Our numerical investigations revealed that a domain-based approach in MPM fails when extremely large plastic deformations are involved The overall performance achieved by the solver is higher than expected and, is even higher with respect to what was reported by Sinaie et al. (2017). We demonstrate that MATLAB is even more efficient than Julia, i.e., the elasto-plastic collapse. This can be avoided when a domain update based on the determinant of the deformation gradient is chosen 
[revised manuscript text omitted]